# Internal climate variability and projected future regional steric and dynamic sea level rise

Aixue Hu [1] & Susan C. Bates[1]

Observational evidence points to a warming global climate accompanied by rising sea levels which impose significant impacts on island and coastal communities. Studies suggest that internal climate processes can modulate projected future sea level rise (SLR) regionally. It is not clear whether this modulation depends on the future climate pathways. Here, by analyzing two sets of ensemble simulations from a climate model, we investigate the potential reduction of SLR, as a result of steric and dynamic oceanographic affects alone, achieved by following a lower emission scenario instead of business-as-usual one over the twenty-first century and how it may be modulated regionally by internal climate variability. Results show almost no statistically significant difference in steric and dynamic SLR on both global and regional scales in the near-term between the two scenarios, but statistically significant SLR reduction for the global mean and many regions later in the century (2061–2080). However, there are regions where the reduction is insignificant, such as the Philippines and west of Australia, that are associated with ocean dynamics and intensified internal variability due to external forcing.

---

[1] Climate and Global Dynamics Laboratory, National Center for Atmospheric Research, Boulder, CO 80307, USA. Correspondence and requests for materials should be addressed to A.H. (email: ahu@ucar.edu)

As the observed global mean temperature increased[1] over the twentieth century, the global mean sea level rose with a mean rate of 1.8 cm/decade[2,3], and increased to 3.1 cm/decade since 1992[4–6] with significant regional heterogeneity[7]. Although there was a warming slowdown from 1998 to 2014[8,9] due mostly to internal climate variability[10–12], the rising sea did not show any sign of abatement[13], because of the nearly constant energy imbalance at the top of the atmosphere causing a net energy gain by the Earth climate system of about 1 W/m$^2$ in recent decades[14,15]. The intergovernmental panel on climate change 5th assessment report[2] and other studies[16–20] project a global mean SLR by the end of the twenty-first century ranging from a few tens of centimeters to a meter or more[18–20] with potentially even larger uncertainty on regional scales associated with the choices of emission scenario (representative concentration pathways or RCPs)[2,18–20] and internal climate processes[21,22]. Here we specifically investigate how internal climate processes could modulate regional SLR from steric and dynamic contributions alone and how much of the regional SLR could be reduced if the climate were to follow a lower emission scenario (RCP4.5) instead of a business-as-usual scenario (RCP8.5) and assess how these processes affect uncertainties in the projected regional and global SLR, topics that have not been thoroughly investigated. For this purpose, we use two sets of unique ensemble simulations from the Community Earth System Model version 1 (CESM1)[23–25] with special focus on the steric and dynamic SLR.

The major processes controlling global and regional SLR in nature are: oceanic net mass change related to an increase/decrease of ice sheets and glaciers[26] and groundwater mining and dam building[27] (mass component), glacial isostatic adjustment (defined as the vertical movement of the earth's crust due to ice sheet mass changes)[2], changes of ocean water temperature (thermosteric) and salinity (halosteric)[2], changes of the earth's gravitational field related to the melting of ice sheets[2], and also ocean dynamics associated with variations of wind-driven or buoyancy-driven ocean circulation[21,22]. Global mean SLR is mostly determined by ocean mass changes and the steric contribution, while regional heterogeneous SLR is mostly related to changes in Earth's gravitational field and ocean dynamics. Neither of these processes change the global mean sea level; instead, they only redistribute the water mass, including heat and salt, within the ocean[22]. For example, as the currents and mass within the ocean shift, sea level rises (SLRs) in one area while falling in another, leading to an uneven change of local sea level[16,17,28–31]. The mass loss from ice sheets reduces the gravitational force between the ice sheets and surrounding ocean water, and also induces a rebounding of the land[2,31], resulting in a higher than the global mean SLR away from melting ice sheets, but a lower than global mean SLR around the melting ice sheets.

Uncertainties for projecting the future regional and global SLR result from uncertainty in RCP scenario assumptions, the uncertain response of ice sheets and the associated land response and gravitational force changes, differences in model physics and configuration, and internal climate processes. As our understanding of the climate system improves, the uncertainties from model physics and configuration, and the response of ice sheets to climate change will dramatically decrease. Although the projected contribution from ice sheets and glaciers on SLR in the future will become much more important, whether the SLR uncertainties from the internal climate processes will change is still unknown, especially on decadal timescales. Since the existence of these internal climate processes is not model dependent and they are physical processes naturally occurring in the climate system, it is extremely important to study how internal climate variability will modulate the projected regional SLR from steric and dynamic contributions.

Here we specifically investigate the internal variability associated with the Atlantic Meridional Overturning Circulation (AMOC), the Antarctic Circumpolar Current (ACC), the North Atlantic Oscillation (NAO), and the Pacific Decadal Oscillation (PDO) and their influence on SLR in the near term (2021–2040) and long term (2061–2080) if the climate pathway were mitigated from RCP8.5 to RCP4.5, assuming that we are currently on the RCP8.5 path. Our results suggest that the changes of the climate pathway can reduce the overall global mean SLR, but the effects on regional SLR due to steric and ocean dynamics are heterogeneous, with significant reductions of SLR in some regions and insignificant ones in other regions even towards the end of the twenty-first century, due to the influence of the unpredictable internal climate processes. It is worth emphasizing again that the purpose of this paper is not to assess SLR due to all processes mentioned earlier but the portion due directly to steric and ocean dynamics, and indirectly through processes associated with coupled internal variability.

## Results

**Global mean and regional SLR**. The simulated global mean surface temperature (GMST) averaged over 2061–2080 increases by 3.05 ± 0.04 °C for RCP8.5 and 1.84 ± 0.03 °C relative to the mean of 1986–2005 (Supplementary Fig. 1a). The corresponding global mean SLR for the same periods due to the steric component is about 17.78 ± 0.15 cm for RCP8.5 and 13.16 ± 0.18 cm for RCP4.5 (Supplementary Data 1 and Supplementary Fig. 1b), which is comparable to the projected global mean steric SLR by the models participating the coupled model intercomparison project phase 5 (CMIP5)[2]. The ensemble mean global decadal steric SLR trend from 2006 to 2080 is 2.82 ± 0.03 cm/decade for RCP8.5, but only 1.99 ± 0.03 cm/decade for RCP4.5 (Supplementary Fig. 2, Supplementary Data 2). Overall, by following RCP4.5 instead of RCP8.5, the decadal mean global warming could be reduced by about 1.2 °C (or 39 ± 1%), and the global mean SLR (decadal trend) reduced by ~26 ± 2% (~29 ± 1%) by 2080. The smaller percentage of reduction in SLR and its trend relative to GMST (26% vs. 39%, and 29% vs. 39%) is related to larger inertia in the ocean compared to the atmosphere, causing the former to take much longer time to equilibrate under increased greenhouse forcing. In other words, even if greenhouse gas emission stops now, the global mean sea level will continue to rise for many centuries[32–35].

On the other hand, the rate of the GMST and global mean steric SLR changes is more significant than their respective mean changes as indicated by a previous study[36]. The mean rate of GMST change in the late twentieth century in CESM1 ensemble (1986–2005) is 0.19 ± 0.14 °C/decade, increasing to 0.39 ± 0.14 °C/decade for RCP8.5 and 0.26 ± 0.13 °C/decade for RCP4.5 in 2021–2040, and to 0.54 ± 0.14 °C/decade for RCP8.5 and 0.22 ± 0.12 °C/decade for RCP4.5 in 2061–2080, suggesting that the rate of GMST changes slows down later in the century for the lower emission scenario with continuous increase for the high emission scenario (Supplementary Fig. 1c). Similarly, the rate of the global mean steric SLR increases from 0.63 ± 0.18 cm/decade during 1986–2005 to 2.16 ± 0.19 cm/decade for RCP8.5 and 1.76 ± 0.18 cm/decade for RCP4.5 in 2021–2040, and to 3.95 ± 0.18 cm/decade for RCP8.5 and 2.33 ± 0.16 cm/decade for RCP4.5 in 2061–2080, respectively (Supplementary Fig. 1d). For RCP8.5, with unabated GMST increasing rate, the rate of global mean steric SLR is nearly doubled in 2016–2080 relative to that in 2021–2040. Although the rate of GMST changes decreases in RCP4.5 towards the end of twenty-first century, the rate of global steric SLR change increases continuously, which reinforces the point that reducing the greenhouse gas emission

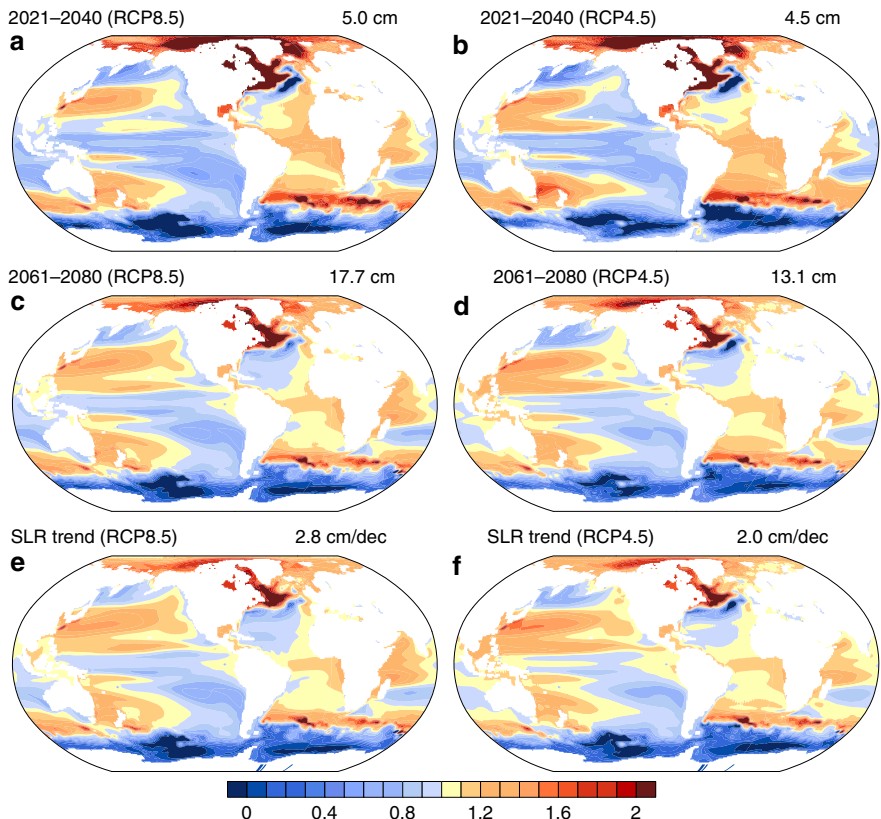

**Fig. 1** Ratio of ensemble averaged 20-year mean sea level rise and the decadal trend of sea level rise and the global mean. **a–d** The mean sea level rise and **e**, **f** the decadal sea level rise trend. The numbers on the top right of each panel are the ensemble global mean value, and shading in all panels have been divided by those global mean values which leads to a unitless shading pattern. For **a–d**, the ensemble global mean sea level rise is relative to the ensemble mean sea level of 1986–2005. The decadal trend is the average 10-year trend over the period 2006–2080. The left panels are for RCP8.5 and right panels for RCP4.5

would not result in an immediate reduction in global mean SLR[34,35].

Regionally, SLR and its long-term trends due to steric and dynamic contributions can deviate significantly from the global mean and between different ensemble members (Fig. 1, Supplementary Figs. 2 and 3)[2,17,19,22]. In general, both the pattern of the ensemble mean mid-century and late-century regional SLR and the pattern of long-term (2006–2080) trends are similar between the two scenarios. Specifically, higher than the global mean SLR is present in the subtropical Pacific, South Atlantic, Arctic, part of the subpolar North Atlantic, and part of Indian Ocean, while lower than global mean is present in the Southern Ocean, subtropical North Atlantic, equatorial Pacific, southeast part of the South Pacific, and subpolar North Pacific in both scenarios. The similarity in pattern suggests that the underlying governing internal processes are similar for both scenarios and over these timescales, and that the ensemble mean SLR could be scaled by the strength of the greenhouse gas forcing[37].

Additionally, the intra-ensemble variability (a measure of internal variability influence) of SLR and decadal trends due to steric and dynamic contributions is larger in regions where the ensemble mean SLR and decadal trends are higher than the global mean, except in the South Atlantic (compare Supplementary Fig. 4 with Fig. 1). In other words, there is less certainty in the SLR projections in the regions that might experience the largest SLR. These variations in regional SLR can generate significant challenges for assessing the potential SLR impact on coastal regions and island nations in the future for any given global mean SLR. For example, with an SLR about twice as much as the global mean, the cities along northeast coast of the United States (such as New York and Boston) need significantly more effort to mitigate the impact of the same global mean SLR than the cities like San Francisco and Buenos Aires, whose SLR is about 15 to 25% less than the global mean (Supplementary Data 1).

In comparison with the late twentieth century, the steric and dynamic SLR variability in the twenty-first century for both scenarios increases in most regions and SLR variance increases more in the long term in RCP8.5 than in RCP4.5, with the main difference in the South Atlantic and Pacific basins (Supplementary Fig. 5). In contrast to the near term, SLR variance in the long term shows an increase in the Indian Ocean, west of Australia, the west and east coastal regions of the Pacific, western coast of Europe, and the Atlantic sector of the Southern Oceans, but a decrease in many other regions for both scenarios (Supplementary Fig. 6a, b). The intercomparison between RCP8.5 and RCP4.5 shows a consistently larger SLR variance in the subtropical north Pacific for both the near term and long term with less consistent patterns for other regions (Supplementary Figure 6c, d). These features may point to a relationship between the forcing scenario and SLR variability indicating that the larger forcing scenario results in larger SLR variability. However, Supplementary Fig. 6c, d shows that the increase in SLR variability lessens with time, which may point to the fact that the forced signal becomes more dominant over internal variability in both scenarios later in the century[38].

For selected global coastal cities, the mean steric and dynamic SLR averaged over 2021–2040 relative to 1986–2005 is not significantly different between the RCP8.5 and RCP4.5 scenarios

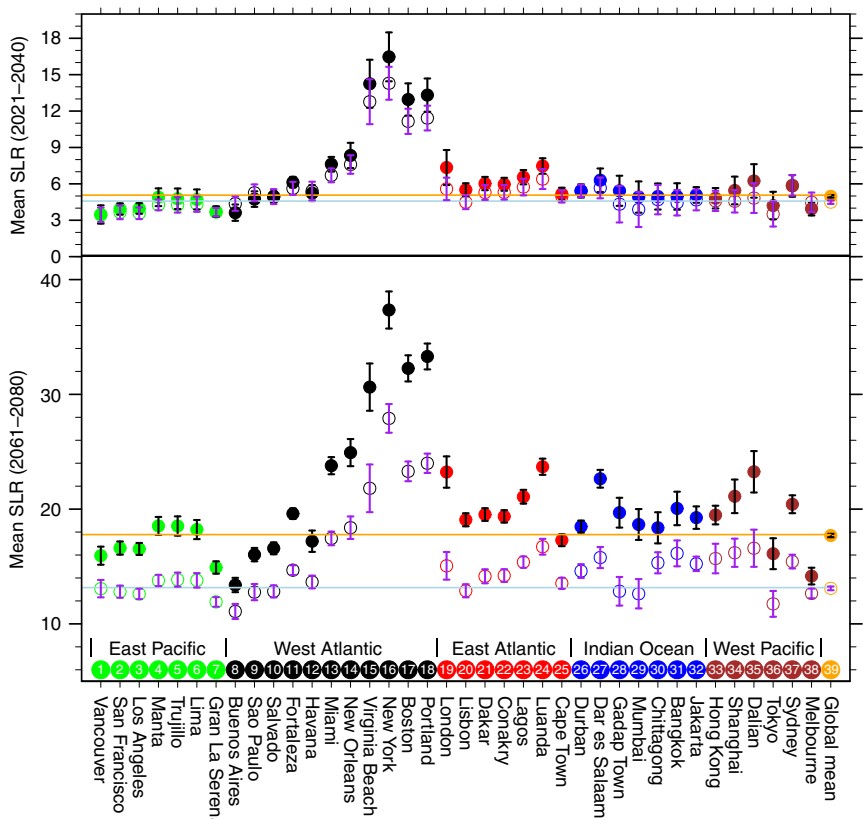

**Fig. 2** Twenty-year mean sea level rise for selected cities. This mean sea level rise is relative to the mean of the 1986–2005. The top is for the mean of 2021–2040 and the bottom for 2061–2080. The geographic location of these cities is given in Supplementary Fig. 7. The solid dots are the ensemble mean sea level rise for RCP8.5 and open circles the ensemble mean sea level rise for RCP4.5. The bars indicate ensemble variability (±1 standard deviation). The units are in centimeters. The brown and light blue line represent the ensemble global mean SLR for RCP8.5 and RCP4.5, respectively. The color coded dots/open circles represent east (green) and west (brown) Pacific coasts, west (black) and east (red) Atlantic coasts, and Indian Ocean coast (blue)

(Fig. 2) due to the internal variability within each ensemble. However, the ensemble mean SLR in the two scenarios averaged over 2061–2080 in most of these selected cities is statistically different. The SLR reduction in this period due to the change of the climate pathway from RCP8.5 to RCP4.5 varies from a few centimeters (such as Vancouver) to over 10 cm (such as New York).

Further analysis on the time evolution of steric and dynamic SLR variability in comparison with SLR mean variability of 1986–2005 shows an overall increase in variability for the global mean SLR in both scenarios (Fig. 3). Within the selected cities, large-scale patterns emerge with lessening of variability along the northeast coast of the United States, the equatorial west coast of Africa, and the west coast of India, but increased variability in nearly all other cities. It may seem counter to the conclusions made earlier that regions with the highest projected SLR have higher variance, specifically in reference to the northeast coast of the United States. This is associated with a reduced AMOC variability which will be discussed later. The consistent changes of SLR variability in both scenarios suggest that the influence of external forcing on internal climate processes is similar under both scenarios. This plot also shows significant decadal timescale variability in the SLR variance, reflecting the influence of decadal scale internal variability. When the internal variability is in-phase for intra-ensemble members, SLR variability is reduced and vice versa for out-of-phase.

On average, the ensemble mean steric and dynamic SLR decadal trends from 2006 to 2080 in RCP4.5 are 20 to 50% less

than that in RCP8.5 everywhere except part of the Southern Ocean where sea level is falling, and the reduction in trends in RCP4.5 compared to RCP8.5 are higher in the subpolar oceans than in the tropical and subtropical oceans (Fig. 4a). Associated with internal climate variability, the regional maximum SLR decadal trends in the RCP4.5 ensemble can be 10 to 40% higher than the minimum SLR trends in the RCP8.5 ensemble, mostly in the western Pacific, eastern Indian Ocean, parts of the subtropical and subpolar North Atlantic, and the Southern Ocean (Fig. 4b). In these regions, the SLR trends are larger in as many as over 50% of the ensemble members in RCP4.5 than the minimum SLR trends in RCP8.5 (Fig. 4c). Therefore, changing to a lower emission scenario could have a smaller effect due to internal variability in cities like Chittagong, Melbourne, Bangkok, and Jakarta, where the model spread of SLR and decadal trends in RCP8.5 ensemble overlap with those in RCP4.5, although the ensemble mean trends are different (Fig. 2, Supplementary Figs. 2, 3, Supplementary Data 1, 2). In other cities, such as New York, Boston, and London, there is no overlapping at all between the spread of the SLR and its trends in RCP8.5 and RCP4.5 giving more certainty to the projection. On the other hand, in comparison with the RCP8.5 ensemble mean SLR trends, the SLR trends of the individual members in RCP4.5 ensemble are smaller almost everywhere, except in the Southern Ocean and parts of the Australian and Philippine coasts where the SLR trend can be higher than the ensemble mean SLR trend in RCP8.5 (6–15% chance, Fig. 4d).

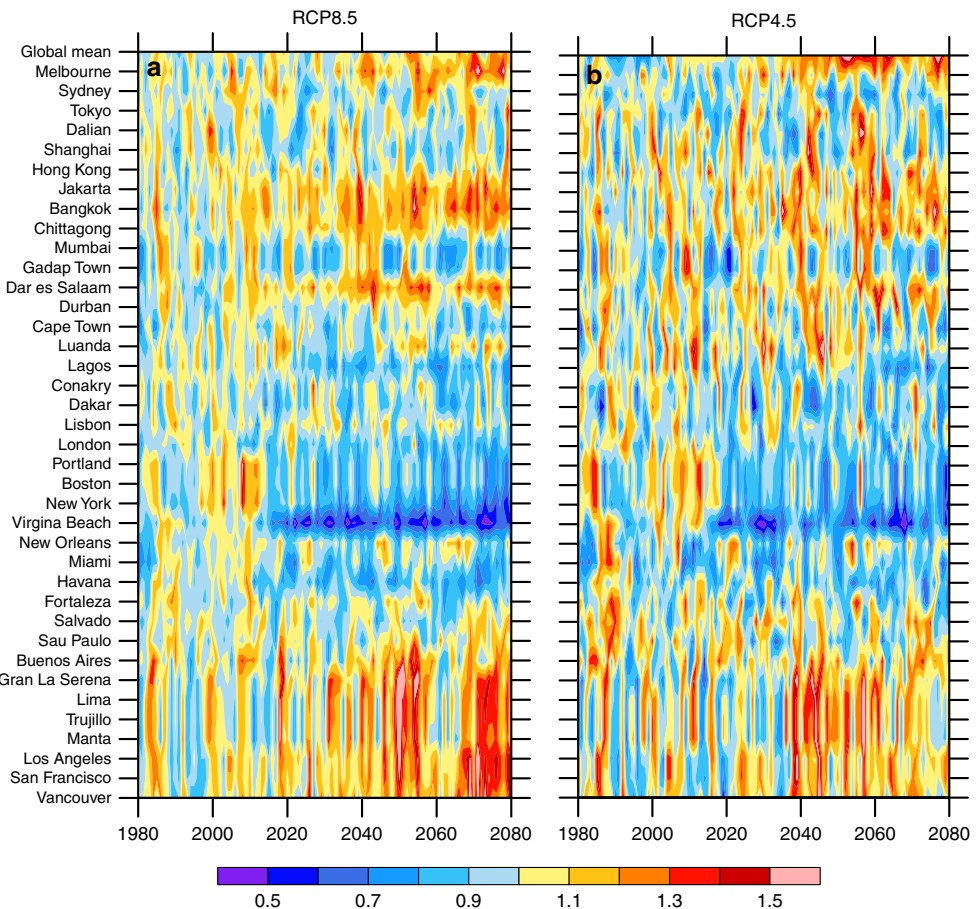

**Fig. 3** Ratio between annual mean SLR standard deviation in selected global cities and the decadal mean SLR standard deviation of the late twentieth century. The SLR standard deviation is defined as the intra-ensemble SLR variance for each of the ensemble simulations. **a**, RCP8.5 ensemble and **b**, RCP4.5 ensemble. Because the figure shown here is the ratios of SLR standard deviations, it is unitless. The late twentieth century is defined as 1986–2005

To get a better picture of how the internal climate processes can modulate the regional steric and dynamic SLR, we compare the potential SLR reduction resulting from following RCP4.5 rather than RCP8.5 with the SLR variability to assess significance. As shown in Fig. 5, the ensemble mean SLR reduction in the near term is not significant at 95% level almost globally, and significant at 68% level only in the eastern Atlantic and subpolar North Atlantic, also including portions of the Southern Ocean and a small part of the eastern subtropical North Pacific. By 2061–2080, the SLR reduction is statistically significant at the 95% level almost everywhere in the global ocean. Insignificant SLR reductions are found in regions west of Australia, South China Sea and east of the Philippines, part of the equatorial western Pacific, and a small portion of the subtropical and subpolar North Atlantic. This comparison further proves that even towards the end of the twenty-first century, the SLR reduction may not be statistically significant in certain regions making it more difficult to plan and prepare in those regions.

**Drivers of ocean variability and change**. As shown in a previous study, the spatial pattern of SLR decadal trends is related mostly to dynamic SLR since the SLR due to thermal expansion of seawater is added as a global uniform number to the dynamic SLR[33]. Since the climate forcing within each ensemble is identical and the only difference is the small perturbation in the initial condition, the variability of dynamic SLR among different ensemble members (Supplementary Fig. 4) is caused only by internal climate processes, such as changes of the wind/buoyancy-driven

ocean circulations, the coupled ocean-atmospheric modes, and the upper ocean stratification. We focus next on exploring some of the main drivers of climate and ocean variability, namely the AMOC, NAO, PDO, and ACC to explain some of the large-scale patterns of SLR and SLR variability noted in the previous section. As shown in previous studies, these internal climate processes are simulated reasonably well in CESM1 in comparison to observations[39–42].

The AMOC is part of a global scale ocean circulation that transports surface warm and salty water to the subpolar North Atlantic where this water loses heat to the overlying atmosphere, becomes dense and sinks to depth, and then flows southward and upwells elsewhere in the world ocean. In both RCP scenarios (Supplementary Figs. 8 and 9), the AMOC weakens with more weakening in RCP8.5 due to the enhanced warming and freshening associated with reduced evaporation minus precipitation (Supplementary Figs. 10 and 11) and the increased exported sea ice meltwater from the Arctic[43,44], which occurs more strongly in RCP8.5. This weaker AMOC leads to reduced northward transport of upper ocean water and a less vigorous Gulf Stream. The latter weakens the surface pressure gradients, and relaxes the slope of the sea surface such that sea level rises north and west of the Gulf Stream and falls in the interior of the subtropical gyre[45,46]. In addition, this weaker AMOC pulls less heat from the South Atlantic to North Atlantic[45], leading to a rise of ensemble mean sea level in the tropical and subtropical South Atlantic and a fall in the subtropical North Atlantic[23,27,47]. In the subpolar North Atlantic, the high rate of ensemble mean SLR there is due to enhanced warming and freshening in this

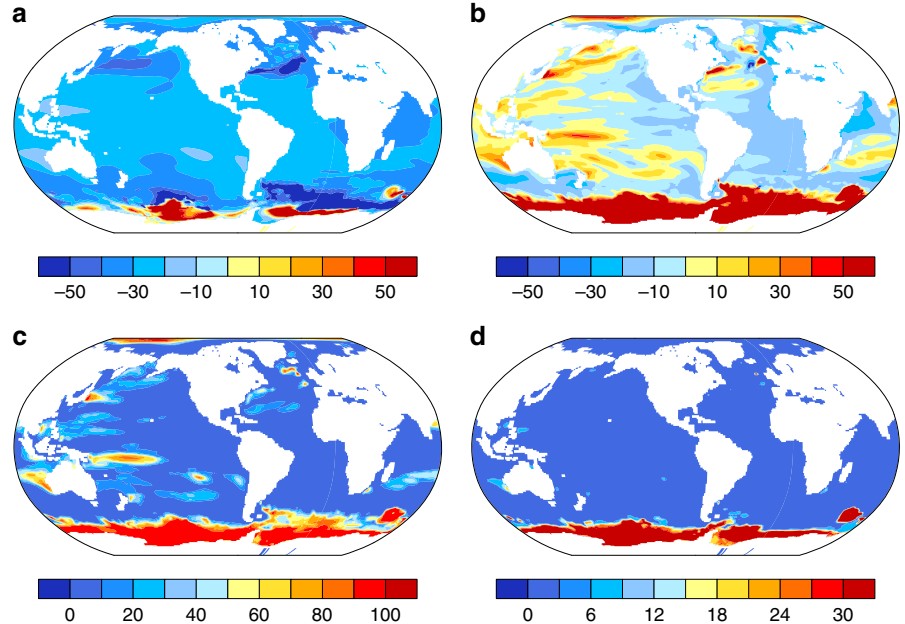

**Fig. 4** Ratio of the SLR decadal trend between RCP4.5 and RCP8.5 ensembles. **a** The ratio between the ensemble mean SLR decadal trend in RCP4.5 and RCP8.5 calculated as the difference (RCP4.5 minus RCP8.5) relative to the RCP8.5 value. **b** The ratio between the maximum SLR decadal trend at each grid point in RCP4.5 ensemble and the minimum SLR decadal trend at each grid point in RCP8.5 ensemble. **c** The percentage chance that the maximum SLR trend at each grid point in RCP4.5 ensemble is greater than the minimum SLR trend at each grid point in RCP8.5 ensemble. **d** The percentage chance that the maximum SLR trend at each grid point in RCP4.5 ensemble is greater than the ensemble mean SLR decadal trend in RCP8.5 ensemble

region and a weaker AMOC. The patterns noted here in relation to a warming climate of decreased AMOC and subsequent impacts to SLR in the Gulf Stream region corroborate results from previous studies[17,36,45–48].

A major source of coupled variability influencing the North Atlantic Ocean is the NAO[49]. Investigation of the sea level pressure pattern, time series, and spectra associated with the NAO for both RCP4.5 and RCP8.5 ensemble members compared to the twentieth century indicates no robust change in magnitude, position, or timescale (not shown). Moreover, the significantly smaller magnitude of the ensemble mean NAO index relative to that in individual ensemble members suggests that the NAO is purely an internal climate process and is not significantly modulated by the changes of the external forcing in either scenario (Supplementary Fig. 12). A comparison of the NAO in RCP4.5 and RCP8.5 reveals no obvious NAO trend in the former, but a small trend towards more positive in the latter, such that the ensemble mean trend is about 0.4 standard deviation over 75 years. Earlier studies show that a long-term positive NAO strengthens the AMOC[50,51], suggesting that the AMOC in RCP8.5 could have weakened more if there had been no upward NAO trend.

On shorter timescales, the interannual variability of the NAO can induce differing strengths of convection in the subpolar North Atlantic[17], affecting the water mass density and the associated oceanic currents, thus further inducing increased regional SLR variability there. From Fig. 6, we see that NAO variability increases only slightly in the twenty-first century compared to the twentieth century. Due to greenhouse gas-induced warming, ocean stratification increases[52], causing a thinner wind-driven circulation layer, resulting in a stronger subpolar gyre, even without the changes of NAO-related wind and buoyancy forcing, leading to an increased variability of the ocean circulation and an increased SLR variability in the twenty-first century. At the same time, the intra-ensemble AMOC variance is significantly less in both future scenarios than in the

twentieth century (Fig. 6) with an overall reduction of variance by about 31%. Therefore, within the CESM, the SLR variability noted in the subpolar North Atlantic (Supplementary Fig. 4) in both scenarios is closely tied to the interannual variability of the NAO in association with stronger oceanic stratification. The reduced intra-ensemble SLR variance is related to the reduced AMOC variance, such as in the Atlantic rim cities (Fig. 3).

To investigate drivers of steric and dynamic sea level changes in the Southern Ocean, we analyze sea level pressure, surface wind, and the ACC (Supplementary Figs. 13 and 14). The ACC strengthens with increasing greenhouse gas forcing in both RCP8.5 and RCP4.5 (but with an initial slow down for the latter) due to a strengthening of the westerly winds, agreeing with previous studies[53–55]. (As shown in Supplementary Fig. 14, a comparison with the CESM1 simulation for CMIP5 shows that the ACC speeds up in both RCP8.5 and 4.5 scenarios. Further analysis shows that the ozone forcing used in RCP4.5 ensemble is derived from different sources which are not consistent with those used in the twentieth century and RCP8.5 ensembles. This difference in ozone forcing induces the initial slowdown of the ACC in the RCP4.5 ensemble.) The change in wind speed is driven by a sea level pressure deepening south of the ACC and rising in the Southern Hemisphere mid-latitudes north of the ACC in both future scenarios. Due to the Coriolis Force, this sea level pressure change drives stronger westerlies. As with the Gulf Stream, geostrophic balance requires a stronger ACC that leads to a larger gradient in sea surface height across the current, causing the ensemble mean sea level to fall south of ACC and to rise north of ACC[53] (Fig. 1). Because the ACC is an eddy rich region, inclusion of realistically simulated eddy effects may modulate our results slightly as suggested by previous studies[53,55].

Another process potentially related to lower rates of ensemble mean SLR south of the ACC is the lower rate of warming experienced by this region in comparison to north of the ACC under increased greenhouse gas forcing[53,54,56]. As shown in Supplementary Fig. 15, the rate of ocean warming in both

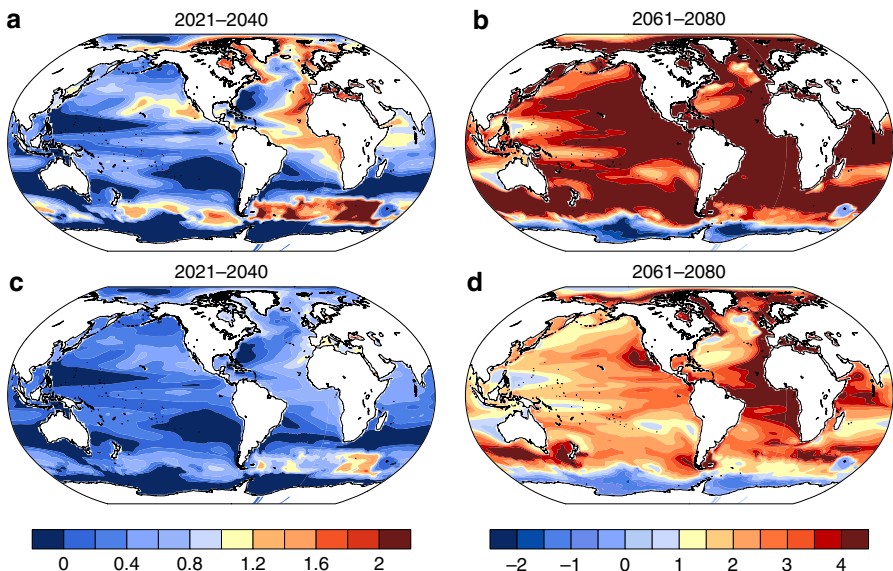

**Fig. 5** Ratio of the sea level rise reduction and the standard deviation intra-ensemble members. **a**, **b** The quotient of the SLR reduction for RCP4.5 from RCP8.5 divided by one standard deviation ($1\sigma$) of the regional SLR within RCP8.5 ensemble for the period 2021–2040 and 2061–2080, respectively, which represents the 68% significance level if the quotient is >1. **c**, **d** The same as in **a**, **b**, but the SLR reduction is divided by $2\sigma$, which represents a 95% significance level if the quotient is >1. The left color bar is for **a**, **c**, and the right color bar is for **b**, **d**

ensemble simulations is higher north of the ACC than south of the ACC[53,54]. Corresponding to this differentiated warming rate, the thermal expansion of seawater would induce more SLR north of the ACC than south of the ACC. These different warming rates are caused by strengthened westerlies which leads to stronger Ekman transport from south of the ACC to north of the ACC. As a result, the upwelling south of the ACC becomes stronger, more cold subsurface water is brought to the surface, and slows the warming there. Thus, the changes of the ensemble mean dynamic sea level in the Southern Ocean are related to two processes—a faster ACC and a differentiated warming on the south and north side of the ACC, both of which are caused by the strengthening of the westerlies.

ACC variability in the two future scenarios also increases in comparison to the twentieth century (not shown), but is not statistically significant. Nevertheless, the SLR variability in the Southern Ocean is larger in these two future scenarios than in the twentieth century, and is larger in RCP8.5 than in RCP4.5 in significant portions of the Southern Ocean. Thus, the increased ACC variability at least partially contributes to the large uncertainty in projecting the Southern Ocean SLR.

As with the NAO in the North Atlantic, the PDO is the leading mode of coupled variability in the North Pacific Ocean[57] and has been tied to sea level changes along the east coast of Asia[58] as well as across the Pacific Basin[59–61]. Similar to the NAO, the PDO is also a pure internal climate mode causing the ensemble mean PDO index to be nearly zero[42]. However, the PDO does undergo a long-term trend, tending toward the warmer phase over the twenty-first century in both future scenarios for all ensemble members (Supplementary Fig. 16). This trend is possibly related to the above global mean warming trend in the North Pacific (Supplementary Fig. 10)[62]. Since the phase and amplitude of the PDO in each ensemble member is different from one to another, the PDO generates similar SLR variability in the Pacific as the NAO in the North Atlantic.

Associated with the PDO trend, the ensemble mean sea level pressure rises in the subtropical Pacific and falls in the subpolar Pacific (Supplementary Fig. 13), leading to a strengthened westerly in the subtropical to subpolar Pacific, and a weakened

easterly in the equatorial Pacific. This wind pattern change causes a strengthened Ekman divergence and upwelling in the subpolar region, leading to a dynamic sea level falling trend, thus a less than global mean SLR in that region, and a strengthened Ekman convergence and downwelling in the subtropics, inducing a dynamic SLR, thus a higher than global mean SLR in that region. In the equatorial Pacific, the weakened easterlies pile less water in the warm pool region, causing a smaller SLR trend in the west equatorial Pacific than the east equatorial Pacific. This pattern of ensemble mean sea level change corroborates the findings of previous studies[22,59,60,63], indicating that the ensemble mean SLR pattern in the Pacific is dominated by changes in the wind forcing associated with the PDO[22].

The PDO variability increases slightly in both future scenarios with a bit larger increase in RCP8.5 (Fig. 6). Associated with the PDO, the variability of subtropical cells (STCs) in the Pacific increases significantly at the 99% level based on Student's $t$ test, although the STCs strength weakens (Supplementary Fig. 9). This larger STC variability strengthens variability of the equatorial upwelling and the subtropical Pacific downwelling in association with the PDO-related wind changes. This compounded with an overall strengthened upper ocean stratification due to greenhouse-induced warming[52] results in larger variability of SLR in these regions (Supplementary Figs. 5 and 6), such as the increased SLR variability in cities along the Pacific rim (Fig. 3). These variability changes are slightly larger in RCP8.5 than in RCP4.5. Although these differences are not statistically significant, they do contribute to the larger SLR variability in RCP8.5 in the Pacific.

In summary, the ensemble mean steric and dynamic SLR and decadal trend are related to the internal variability as discussed above. A weakened AMOC induces an increased (decreased) dynamical SLR north/west (south/east) of the Gulf Stream as well as an increase in the tropical and subtropical south Atlantic and decrease in the subtropical north Atlantic. The small trend toward a more positive NAO in the RCP8.5 may enhance the increased (decreased) dynamical SLR noted in the subpolar (subtropical) gyre. Although the Atlantic basin is dominated by the AMOC response, the Pacific basin pattern is consistent with

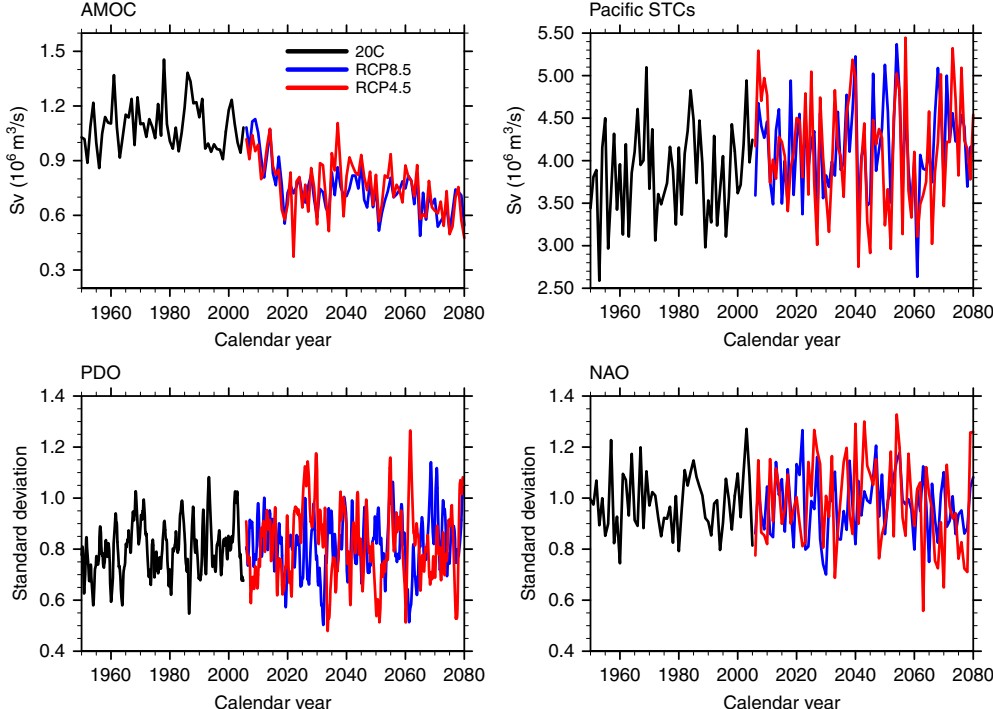

**Fig. 6** Time evolving intra-ensemble standard deviation for a given internal climate processes. AMOC Atlantic Meridional Overturning Circulation, Pacific STCs Pacific Subtropical Cells, PDO Pacific Decadal Oscillation, and NAO North Atlantic Oscillation. Black lines represent the intra-ensemble standard deviation for the twentieth century (1920–2005); blue lines are the intra-ensemble standard deviation for RCP8.5 (2006–2080); and red lines the intra-ensemble standard deviation for RCP4.5 (2006–2080)

the positively trending PDO via associated wind changes, indicating a higher dynamical SLR within the subtropical gyres (Fig. 1). Consistent with weaker easterlies in the equatorial Pacific, SLR is less in the western region and higher in the east, most notably in the latter part of the century (Fig. 1c, d). A decrease (increase) in SLR south (north) of the ACC over the twenty-first century (Fig. 1e, f) matches the expected pattern associated with a strengthening ACC. The nearly global enhancement of SLR variability (Supplementary Fig. 5) may be related to the overall shallowing of the mixed layer due to increased stratification causing a faster and larger response of upper ocean circulation due to wind stress fluctuations[52].

Our results further show an increase in variability of the PDO and STCs in the Pacific and the ACC in the Southern Ocean (especially in RCP8.5), which contributes to the overall larger SLR variability and less significant SLR reduction (difference between RCP8.5 and RCP4.5) in these regions. The AMOC variability lessens in the twenty-first century relative to twentieth century and the variability is smaller in RCP8.5 than in RCP4.5, which contributes to less SLR variability in most parts of the Atlantic and a more significant SLR reduction, especially in the long term due to a much larger weakening of the AMOC in RCP8.5. The variability of the NAO does not have long-term changes in our simulations for both scenarios, but its interannual variability does contribute to the intra-ensemble SLR variability in the North Atlantic.

## Discussion

Internal climate variability can play a key role in determining the projected regional SLR, especially in the near term. By following a lower emission scenario instead of business-as-usual scenario, the ensemble mean steric and dynamic SLR could be reduced by 26 ± 2% for the global mean by 2061–2080 as suggested by two sets of

ensemble simulations using CESM1. Regionally, the ensemble mean SLR reduction due to changes in future climate pathway varies from ~20 to 50%. The ensemble mean SLR reduction is statistically insignificant in the near term both globally and regionally because there is little difference in the projected greenhouse concentration level for these two scenarios in this period and internal climate processes have a large influence. Toward the end of the century, SLR projections are well separated indicating that external forcing is dominating and internal variability is playing less of a role. Therefore, any policy on limiting greenhouse gas emission will not immediately reduce the overall impact of climate change on low-lying coastal regions and island nations due to the rising sea in the near term, but will on multidecadal or longer timescales.

Moreover, due to the non-uniform pattern of regional SLR associated with ocean dynamics, the ensemble mean SLR reduction may not be statistically significant even in the long term in regions such as Australia and the Philippines due to enhanced internal climate variability, but can be very significant in other regions such as the east coast of North America, which exhibit lessened internal climate variability. Broadly speaking, regions with higher variance, and thus reduced certainty, coincide with mid-latitude wind-driven circulation, such as western boundary currents, subpolar and subtropical gyres, and the Southern Ocean. Exceptions to this, such as the east coast of the United States and South Atlantic, which respond mainly to long-term changes of the AMOC, are mainly due to trends in regional driving mechanisms or regions/time periods where the forced signal overcomes internal variability. Consequently, more certain projections and thus better planning can be made in some regions while not in others.

It is worth noting that all ensemble simulations used here are branched from the same twentieth century all-forcing simulations on January 1, 1920 with round-off level perturbation in the

atmospheric initial field with identical ocean, sea ice, and land initial conditions using the same version of the CESM model. All uncertainty sampled here is related to the small perturbation of the atmospheric initial condition and the interaction between the atmosphere and the other components. Although changes of the external forcing may modulate internal variability, this effect is similar in our simulations within each ensemble since the same model and identical external forcing are used. A recent study shows that when the initial conditions of the ocean are also perturbed, the uncertainty seen here increases significantly in certain regions[64]. Moreover, since only the steric and dynamic SLR have been examined here, the SLR uncertainty sampled here may be at the lower bound of the real SLR uncertainty. Last, if multi-models are used, the effect of the internal variability on regional SLR will be severely contaminated by the differences in model physics and the horizontal–vertical configuration. This could result in larger uncertainty in projected SLR due to the combined effect of the internal climate variability and the model differences. Thus to better isolate the influence of the internal climate variability on projected SLR, a single model with a large ensemble of simulations, as is done within this study, is a better choice. If large ensembles become available from different modeling groups, this study could be repeated with each ensemble and a comparison of SLR due to internal variability among the models can be conducted.

Another source of uncertainty in estimating future SLR is associated with ice sheets, glacial isostatic adjustment, gravitational force, and land water storage and dam building including deep water pumping. Observations and model projections suggest an increased contribution to SLR from ice sheets and glaciers since at least the 1990s[65,66]. With this increased mass contribution to SLR, the influence of glacial isostatic adjustment and gravitational force on regional SLR increases. Because these processes have not been physically coupled with ocean dynamics, the current practice is to obtain the influences of these processes offline, and then linearly add these influences onto the model produced dynamic SLR field[2,67–70]. By doing so, it neglects the potential interaction between the ocean dynamics and these processes, leading to physical inconsistency. With future improvements to model physics, these missing physical processes will eventually be incorporated into the climate models in a decade or two, which will dramatically advance our ability to more adequately estimate projected SLR uncertainty and how this uncertainty will affect our ability to mitigate the greenhouse gas-induced SLR.

## Methods

**Model**. CESM1 is a global scale, fully coupled climate model developed at the National Center for Atmospheric Research in collaboration with scientists in universities and the United States Department of Energy laboratories[23]. The horizontal resolution for all components is nominal one degree with enhanced horizontal resolution in the equatorial regions for the ocean component. CESM1 does not include an active ice sheet model, glacial isostatic adjustment, or gravitational force changes. This limits our ability, allowing only estimation of steric and dynamic SLR, which together accounts for approximately 40% of the observed global mean SLR for recent decades[2,65] and may become even less percentagewise to the total SLR by the end of this century (although the absolute contribution from steric and dynamic SLR increases)[2,19,20]. The SLR due to a net heat gain/loss is calculated by vertical integration of the seawater in situ density due to temperature changes and summed globally[33]. Then, this global mean thermosteric sea level change is added on the dynamic sea level field from the model simulations to obtain the regional SLR due to both ocean dynamic and thermosteric effects. Within this manuscript, unless specifically clarified, SLR discussed only includes the thermosteric and dynamic parts.

**Experiments**. The two sets of ensemble simulations used here start from the same twentieth century all-forcing simulation branched from an 1850 preindustrial control simulation. On January 1, 1920, a round-off level perturbation is introduced into the air temperature field from ensemble member #1 and run to 2005 (30

members in total). After 2005, the simulations follow two different climate pathways—RCP8.5 (30 members) and RCP4.5 (15 members) from 2006 to 2080. The purpose of this large ensemble project was to study the effect of internal climate variability on various climate phenomenon. For details of the experimental design, please see refs. [24,25] for the RCP8.5 ensemble and the RCP4.5 ensemble, respectively. The linear trend from the preindustrial control run is removed from all SLR data. To link the potential impact of the SLR to human societies, here we use the SLR for selected global cities shown in Supplementary Fig. 7 as examples. The SLR for these cities is defined as the SLR of the closest ocean grid point, which may underestimate the actual SLR variability due the unresolved physical processes, such as ocean eddies and the detailed shape of coastlines. Additionally, the SLR noted for specific cities in this work is actually part of a larger, regional response of SLR to changes in the external forcing because the SLR is driven by large-scale climate variability (like the NAO, PDO, etc.); therefore, we can make generalizations about specific cities based on these larger patterns even if we had not done the analysis at specific grid points. In reality, the coastal SLR is not only controlled by large-scale climate patterns but also affected by local winds on a spatial scale of tens kilometers in association with the ocean topography and shape of the coastlines (may also be affected by changes of tides on different timescales). If these processes were included in our simulations, it certainly would induce larger SLR variability in these coastal cities.

**Data availability**. All data used here can be openly accessible through the following links:

The CESM1 Large Ensemble Project homepage: http://www.cesm.ucar.edu/projects/community-projects/LENS/

The model output for the 1850 control run and the twentieth century historical runs and the RCP8.5 simulation can be downloaded from Earth System Grid (ESG) URL:

https://www.earthsystemgrid.org/dataset/ucar.cgd.ccsm4.CESM_CAM5_BGC_LE.html

The RCP4.5 ensemble (ME) can be downloaded from this ESG URL:
https://www.earthsystemgrid.org/dataset/ucar.cgd.ccsm4.CESM_CAM5_BGC_ME.html

The post-processed sea level data can be obtained through contacting the corresponding author (ahu@ucar.edu).

All data processing and figures used the NCAR Command Language (Version 6.4.0) [Software]. (2017). Boulder, Colorado: UCAR/NCAR/CISL/TDD. https://doi.org/10.5065/D6WD3XH5.

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

## Acknowledgements

A portion of this study was supported by the Regional and Global Climate Modelling Program (RGCM) of the US Department of Energy's Office of Science (BER), Cooperative Agreement No. DE-FC02-97ER62402. This research used computing resources of the Climate Simulation Laboratory at the National Center for Atmospheric Research (NCAR), which is sponsored by the National Science Foundation. The National Center for Atmospheric Research is sponsored by the National Science Foundation. We thank Warren G. Strand for his help with data processing.

## Author contributions

Both A.H. and S.C.B. have actively participated in data analysis and writing the article.
