## [Peer Review File · Nature Communications]

Reviewer #1

Review of Hu and Bates: "Influence of internal climate variability in mitigating the projected future regional sea level rise"

Summary

In this paper, 2 sets of ensemble simulations from CESM1 (RCP4.5 and RCP8.5) are compared for the sea level projections up to 2080. Initially the goal of the paper is not quite clear, but at some point I thought it would be about comparing the magnitude of internal variability on global and regional SLR in 2 different RCP scenarios. In the first part of the analysis there is a lot of discussion on trends, and in the second part on some internal climate variability phenomena, but in the end I can't really find where the difference in sea level internal variability between the two scenarios is discussed. Perhaps that means I misunderstood the question?

Analysis

I find it quite difficult to pinpoint what is new & noteworthy in this paper, this should be made much clearer early on: what is it that was done here that has not been done before? If this manuscript would be much more focused and more clear in its explanations, it would probably make for an interesting contribution – looking at the internal variability in regional sea level and how it is impacted by the scenario. You have this great large set of realisations branching from the same run, which might be able to help you look at the internal variability – please use it!

However, as this manuscript is now, it seems to be dealing with analyses that are not exactly novel (on regional patterns of the thermal expansion and the ocean dynamics component), which was for instance already discussed in IPCC and several papers on regional sea level projections.

What I find quite confusing is the term 'mitigating from RCP8.5 to RCP4.5', which is repeatedly used. To me this suggests that the climate model initially follows the RCP8.5 scenario, which is (at some point) changed into an RCP4.5 scenario. (eg title, L84-85, L90, L98). But instead this phrase seems to refer to the difference between the two scenarios?

I have some other comments and questions, sorted by section, which I hope will help the authors in preparing a new version of the manuscript. Essentially I think all ingredients are there but the way they are presented could be improved, as well as the choice of which data to focus on in the analysis.

Comments

Title:

- Is the question that is answered really how internal climate variability mitigates the RSL, or is the question actually: how big is the effect of internal climate variability in different climate scenarios?
- It should be made clear early on that this study is looking at the thermal expansion and dynamics only, which is only less than half of the actual SLR.

Abstract:

- It is not clear from the abstract what is new about this research? The reader has to wait until the end of the introduction (L58) to find this out.
- In L15-21, are these numbers comparing the difference between RCP4.5 and RCP8.5 for the thermal expansion or for the internal variability effect (the difference in AR5 is about 8 cm, so it is maybe the latter?)

- Time period is not mentioned (later it turns out to be 2061-2080 relative to 1986-2005, this is quite crucial for comparisons)

Introduction

- It takes a very long time to get to the point, especially since the reader still has no idea where we're heading after the abstract
- L13-14 'the potential benefits of the sea level change' sounds like 'there are also positive sides to SLR', while I suppose the message that is intended is: 'can we get less sea level change if we would follow a lower emission scenario instead'?
- L30 'recent centuries'?
- L35 'mass component' would be more correct. 'eustatic' is not really used anymore in this context as it's actually defined as the sea-level change in a rigid ocean basin (check for instance Rovere et al, 2016, <http://link.springer.com/article/10.1007/s40641-016-0045-7>)
- There is lots of repetition in L27-48; jumping back and forth between dynamics/ice, global/regional – could be shortened
- L45 what is 'melt-back'?
- L49-52 repetition in sentence
- L54 moreover = however
- L58-L61: This is FINALLY explaining what will be investigated! Please state this in the abstract!
- L59 what is 'global mean regional SLR'??
- L61-62 I would take out this sentence, it only confuses rather than clarify - unless the authors actually show a model simulation driven by RCP8.5 for the first half of the century and by RCP4.5 for the second half.

Methods

- L71-72 This is cryptic. Why is this done and what does it mean?
- L73-74 So there are 30 members in the historical run, then 30 go on for RCP8.5 and (the first?) 15 for RCP4.5? Why?
- L75 why 2080, it seems quite random? Using this period makes it also very hard to compare to IPCC (or other sources).
- L77-79 this should be said in the introduction already
- Is there no drift correction applied to the ocean variables?

Global and Regional Mean SLR

- (should this subtitle be 'global mean and regional SLR'?)
- L92: these numbers seem low compared to IPCC (or at least at the lower end), and the uncertainties (unclear if this is 1 sigma??) are very small, why?
- L104: 'in other words'. Suggest change 'global warming' to 'greenhouse gas emissions'?
- L108: strange wording, how can SLR and the SLR trend be similar? (same confusing terms in the rest of this paragraph)

Dynamic Sea Level Mechanisms and Changes

- L164-165 this is true by definition, but only if the thermal expansion and dynamics are the only SLR terms considered. If ice mass changes and other contributions are added, even a place with below-average dynamics can be above the global mean total SLR.
- It would be good to include some discussion on the ability of the climate models to reproduce internal variability in the first place. How good is the magnitude, spatial distribution and timing of the variability in the model?

- L198-200: The model shows a weakening AMOC and a positive NAO, seemingly in contradiction to this statement. Or does the statement in this sentence mean that the AMOC would have been even weaker if it weren't for the strengthening NAO?
- L207 – There are quite some papers that have discussed this pattern in the Southern Ocean due to a shift in the ACC, it would be good to reference at least a couple here (for instance Böning, C. W., A. Dispert, M. Visbeck, S. R. Rintoul, and F. U. Schwarzkopf, 2008: The response of the Antarctic Circumpolar Current to recent climate change. *Nat. Geosci.*, 1, 864–869, doi:[10.1038/ngeo362](https://doi.org/10.1038/ngeo362); Frankcombe, L., P. Spence, A. M. Hogg, M. H. England, and S. M. Griffies, 2013: Sea level changes forced by Southern Ocean winds. *Geophys. Res. Lett.*, 40, 5710–5715, doi:[10.1002/2013GL058104](https://doi.org/10.1002/2013GL058104).)

Summary

- Coming back to my comment reg L58-61, I thought that this was what would be investigated, but somehow by the end of the paper I'm not too sure anymore, because it is barely discussed in the summary.
- It is not that surprising that the differences between RCP4.5 and RCP8.5 are not very big on the short term, because the forcing only really starts to diverge later in the 21st century.
- It should maybe be mentioned somewhere early on that the 'city values' are actually a pretty big grid cell that is close to the location mentioned.

Figures/tables:

- Figure 1: the trend period should be mentioned here
- Figure 2: I find it confusing that the time period leading to larger sea level rise is below the shorter period with less sea level rise
- There is a typo in the caption of SupFig1. What are the uncertainties here, 1 sigma? What are the uncertainties anywhere in the paper?
- SupTable1: caption should be "Global and regional sea level rise in 21st century (cm)". Might it be better to put percentages of the global mean rather than cm?
- How come different places are below average between the two scenarios? (there are more in the RCP85 scenario?) -> it is exactly this that I would have expected to be discussed in this paper, does this have to do with reduced (or enhanced) internal variability?
- SupFig3: a line indicating the global mean would be handy. What is the grey dot at the global mean? Note that in the text this figure is mentioned in text before SupFig2.
- SupFig6: this should probably be done by a scatter plot: the line (and areas) suggests this data is continuous and connected. Also, I have no idea what this graph is supposed to tell me and how it supports the claims made in L132-136?

General assessment

Unfortunately, I cannot recommend publication of the manuscript due to the reasons listed under “major comments”. For the paper to be published at all it would need significantly revised because there are a number of problems in the interpretation of the simulation results, but even after such revisions, I do not see it to rise to the standards in novelty and significance that is generally required within the NPG. However, in particular this question of significance should, of course, be decided by the editor.

Major comments

1. Confusing statements: The title promises a comparative investigation of internal variability and future trend with respect to sea level changes, but already in the abstract (and through-out the paper) the focus lies on the difference in global mean sea level change between different emission scenarios. This comparison has been done short of a million times and is not a high profile result. It took a while for me to understand what the authors mean by “internal variability”: They claim that most of the spatial differences of sea level changes arise from internal variability. I think that is by no means justified because the regional sea level changes will be in balance with the oceanic circulation and the wind stress changes and changes in both are likely to arise from changes in the climate system. The assumption that this is “internal variability” is particularly puzzling because the authors average them over a decade.
2. Missing novelty: The thermal expansion of the ocean has been reported a number of times. Previous studies, for example by the first author, even used ensembles of models. The comparison between different warming scenarios is a standard result and I do not see how the paper provides anything new or special with respect to method, approach or result.
3. Overselling:
 - a. The thermal expansion constitutes only 40% of the global mean sea level rise. The title, abstract and most of the paper are written as if it was equal to the global mean sea level rise. That is not appropriate.
 - b. The paper falls behind previous studies used an ensemble of different climate models for similar analysis by the same lead author. The model differences are significant and it is not clear why the reader is to believe that the results from one model is providing the full answer. It is however presented as the full answer.
4. The literature is not up-to-date. Although there are some references of 2016, most of the literature reflects the scientific state of five years ago. Practically no reference to the work after the latest assessment report of the IPCC and almost no reference to glacier or ice sheet contributions.

5. To provide specific sea level rise values for different cities is problematic if only the thermal expansion of a coarse resolution climate model is provided. The spatial precision that is suggested by selection of a city is not appropriate when the numbers provided (1) are from a coarse resolution model, (2) only cover 40% of the currently observed sea level rise and (3) do not include potential contributions for example from tectonic uplift which can be as strong as the regional sea level or the global sea level rise.
6. The results of the simulation are generally just reported and not explained (for example by ocean circulation changes).

Selected minor comments

1. Literature: The literature seems outdated and at places very imprecise. Some example: To cite the IPCC in line 37 for the thermosteric and halosteric effect is like citing a text book on the Navier-Stokes equation. There is no use to it in a scientific publication. Also the collection of references, for example in lines 58 and 183, is not very helpful and seems to merely serve the purpose of putting a lot of papers in the reference list as opposed to inform the reader about results previously reported in the literature.
2. Fig. 2: The colouring of the dots should be explained in the figure caption.

Reviewer #3

Overview

This paper represents a nice overview of various papers and IPCC reports. Much of the “results” have been published in previous papers: the global projections are in the IPCC and the regional projections are also found in the IPCC and various sea level rise intercomparison papers. This includes the impact of internal variability, although I acknowledge the discussion found in the intercomparison papers is not as clear/direct as in this paper.

The new part is a clear discussion of the impact of mitigation on the regional projections and using many ensemble members of 1 model. BY using the many ensemble member of 1 model the influence of variability is reliably assessed. The papers previous assessed the impact of variability by using a multi-model ensemble although this mixes response differences and variability. This paper represents a clear improvement of the discussion of SLR.

My recommendation is to publish the paper since it is very readable and understandable to most non-specialists. Further it presents some new and interesting results. That said, I have a number of specific comments the authors should address before the paper is published. This are all relatively easy to address. I would not have to see the paper again before publication.

Specific Comments

1. Line 1 – Title – I found the title confusing and/or misleading. A suggestion to change it – Influence of mitigation and internal climate variability on the projections of future regional sea level rise.
2. Lines 9 – global mean sea level rise is also rising – reads funny. Change to . global mean sea level rise also rises.
3. line 17 – RCP – I would argue that “RCP scenario” is jargon. Change to “emission scenario”.
4. Line 19 – marginal – Is “small” better?
5. Line 20 – Delete “very large”. Change to “larger”. 10cm (the difference between sites) of SLR may or may not be large. Also, this assessment of “large” depends on the time scale in view.
6. Line 21 – Shouldn’t the Yin et al. (ref 20) be cited here?
7. Line 26 – by about 1 – Change “by” to “of”.
8. line 27 – Change “this sea level rise” to “the observed sea level rise”. It makes the meaning clearer.
9. Line 42 Change “total heat” global heat”. “Total” is the wrong word here.
10. Lines 43 -45 – Could add reference to Gregory et al. model intercomparison SLR paper to list. 2001: Comparison of results from several AOGCMs for global and regional sea-level change 1900-2100. *Climate Dynamics*, 18(3/4), 225-240.
11. Line 99 – Somewhere near here, it should be noted that the values being discussed are strongly dependent on the time scale of interest. Also, add “by 2080” after “29%”.
12. Line 105 – Could add a reference to Stouffer et al. 1999: Response of a coupled ocean-atmosphere model to increasing atmospheric carbon dioxide: Sensitivity to the rate of increase. *Journal of Climate*, 12(8), 2224-2237.

13. Line 113 – After “for these scenarios” add “and over these time scales”.
14. Line 178 – Could cite any one of a number of old (circa 1990’s) papers which highlight these processes. This is not a new result.
15. Lines 210 – 221 – Should not there be an eddy caveat here?
16. Lines 222- 225 – Again, this is an old result. Please cite the early papers.
17. Line 239 – Compared to other CMIP models or just the ensemble of this model? Not clear.
18. Lines 273 – 281 – Several caveats are missing here.
These runs are missing
 1. land water storage and adding (dams and deep water pumping).
 2. land ice melt/freezing.
 3. gravity changesTherefore, the details will change when model can account for those changes.
19. Figure 1 caption – Line 432 – “top left” should be “top right”.

All panels are weighted by those global mean values – what does this mean? The global mean is removed?

20. Figure 2 caption - - line 443 – variability – Maximum to minimum? ... or what?
21. Figure 3 – What color scale goes with what figure? I could not figure it out. Sorry for the pun. Ratios are displayed according to the caption. I do not see a color to fit this. Are the values percentage change?
22. Figure 4 caption – line 457 – I assume this is a ensemble mean. Correct?

Supplement

23. Table 1 and 2 – Column labels – Use one style – All caps or not. There is a mixture currently.
24. Figure 1 – Very hard to see individual model lines.
25. Figure 4 – Really hard to see anything meaningful. Chose a few panels to highlight.
26. Figure 5 caption/figure – What is SLC? Define.
27. Figure 6 – Green line is very hard to see.
28. Figure 12 – a) and b) is figure not defined in caption.
29. Figure 14 – I think the 20C panel should be on top. The RCP8.5 panel should be the third down from the top.

We sincerely thank the three anonymous reviewers for their constructive and insight comments, we have revised our manuscript accordingly. Our major changes are:

- 1. As suggested by reviewers, we have rewritten the abstract and make it read better and clearer. We also change the abstract to be consistent with the format of Nature Communication.*
- 2. We have rewritten the introduction section to streamline what is new here, what we have done here, why we want to do this, and how we did it.*
- 3. For the result section, we have rearranged our figures. We moved the old Figure 4 to the supplementary material and added three new figures. The three new figures are: Figure 3 to show the time-evolving ratio between annual mean SLR standard deviation in selected global cities and the decadal mean SLR (1986-2005) standard deviation; Figure 5 to show the ratio of the sea level rise reduction and the standard deviation across ensemble members which is a measure how the internal climate variability can influence the potential SLR reduction due to a lower emission scenario being followed; Figure 6 to show the time evolution of the intra-ensemble variance for AMOC, STC, PDO, and NAO.*
- 4. In the result section, we have significantly rewritten this section and added discussion on how the SLR variability changes over time and how this SLR variability is affected by the internal climate processes.*
- 5. For the discussion section, we have rewritten the summary to make our points clearer and also added discussions on the potential impact of the physical processes which have left out of CESM 1.*
- 6. We also updated our reference list to catch up the recent progress in regional and global SLR.*

In the response to reviewers, the original comments are in black and our responses are in blue.

Response to reviewer #1

We thank reviewer #1 for the constructive and insight comments. We have revised our manuscript based on these comments.

In this paper, 2 sets of ensemble simulations from CESM1 (RCP4.5 and RCP8.5) are compared for the sea level projections up to 2080. Initially the goal of the paper is not quite clear, but at some point I thought it would be about comparing the magnitude of internal variability on global and regional SLR in 2 different RCP scenarios. In the first part of the analysis there is a lot of discussion on trends, and in the second part on some internal climate variability phenomena, but in the end I can't really find where the difference in sea level internal variability between the two scenarios is discussed. Perhaps that means I misunderstood the question?

The understanding of this paper by reviewer #1 is correct. In the revision, we have made more discussion on how the sea level internal variability changes and how these changes affect the projected sea level rise reduction if RCP4.5 is followed instead of RCP8.5.

Analysis

I find it quite difficult to pinpoint what is new & noteworthy in this paper, this should be made much clearer early on: what is it that was done here that has not been done before? If this manuscript would be much more focused and more clear in its explanations, it would probably make for an interesting contribution – looking at the internal variability in regional sea level and how it is impacted by the scenario. You have this great large set of realisations branching from the same run, which might be able to help you look at the internal variability – please use it!

Thanks for this comment. In our revision, we have tried to make these points more clear. We have stated the purpose of this study in the first paragraph of the introduction section. In the summary section, we have pointed out more clearly what is new in what we have done.

However, as this manuscript is now, it seems to be dealing with analyses that are not exactly novel (on regional patterns of the thermal expansion and the ocean dynamics component), which was for instance already discussed in IPCC and several papers on regional sea level projections.

Actually what we want to demonstrate here is what global and regional SLR could be avoided and how the internal climate processes induce SLR uncertainty and contribute to regional SLR differences in the future scenarios when a moderate emission scenario (RCP4.5) is followed rather than a business as usual scenario (RCP8.5). We would like to see whether the internal variability induced uncertainty will modulate this SLR difference or not. What we can see is that the internal variability contributes significantly to the uncertainties of the projected sea level rise on regional scales, especially on decadal timescales. For example, the sea level rise reduction in the near term is insignificant in most regions due to the internal processes when we follow a low emission scenario (RCP4.5) instead of a high one (RCP8.5). Although the regional pattern of sea level rise in both RCP4.5 and RCP8.5 has been discussed before, the role of the internal variability on the potential avoided sea level rise has not been specifically discussed.

In our revision, we have tried to make all these points more obvious.

What I find quite confusing is the term ‘mitigating from RCP8.5 to RCP4.5’, which is repeatedly used. To me this suggests that the climate model initially follows the RCP8.5 scenario, which is (at some point) changed into an RCP4.5 scenario. (eg title, L84-85, L90, L98). But instead this phrase seems to refer to the difference between the two scenarios?

This is a nice comment. In our revision, we try not use this term. Our original scope of this paper is to study how mitigation from RCP8.5 to RCP4.5 could reduce the global and regional sea level rise and whether the expected reduced sea level rise will be affected by the internal climate variability. In general, we assume we are following the RCP8.5 scenario now as a business as usual case since there is no serious policy in place to curb the CO2 emission for the countries who are the major CO2 emitters.

I have some other comments and questions, sorted by section, which I hope will help the authors in preparing a new version of the manuscript. Essentially I think all ingredients are there but the way they are presented could be improved, as well as the choice of which data to focus on in the analysis.

Thanks for this comment which made our paper better focused.

Comments

Title:

- Is the question that is answered really how internal climate variability mitigates the RSL, or is the question actually: how big is the effect of internal climate variability in different climate scenarios?

The question answered is how SLR may be reduced on global and regional scales when following the RCP4.5 rather than RCP8.5 and how internal climate variability might modulate SLR on regional scales differently between the two scenarios.

- It should be made clear early on that this study is looking at the thermal expansion and dynamics only, which is only less than half of the actual SLR.

We made this clear in a few places: in the introduction: “Here we specifically investigate how internal climate processes could modulate regional SLR that could be avoided if the climate were to follow a lower emission scenario (RCP4.5) instead of a business-as-usual scenario (RCP8.5) and assess how these processes affect uncertainties in the projected regional and global SLR, topics that have not been thoroughly investigated. For this purpose we use two sets of unique ensemble simulations from the Community Earth System Model version 1 (CESM1)²³⁻²⁵ with special focus on the thermosteric and dynamic SLR.” In the method section:” This limits our ability, allowing only estimation of thermosteric and dynamic SLR, which together accounts for approximately 40% of the observed global mean SLR for recent decades^{2,32} and may become even less by the end of this century^{2,19,20}.”, “Hereafter, unless specifically clarified, SLR discussed in this paper only includes the thermosteric and dynamic parts.” We also added a caveat on the potential influence of other components to regional SLR.

Abstract:

- It is not clear from the abstract what is new about this research? The reader has to wait until the end of the introduction (L58) to find this out.

In the revised abstract, we made this clearer. Here is part of the new abstract: “Here, by analyzing two sets of ensemble simulations from a climate model, we investigate potential SLR that may be avoided if a lower emission scenario is followed instead of business-as-usual one over the 21st Century and how it may be modulated by internal climate variability. Results show almost no statistically significant difference in thermosteric and dynamic SLR on both global and regional scales in the near-term between the two scenarios, but statistically significant SLR reduction for the global mean and many regions later in the century (2061-2080). However, there are regions where the reduction is insignificant, such as the Philippines and west of Australia, associated with ocean dynamics and intensified internal variability due to external forcing.”

- In L15-21, are these numbers comparing the difference between RCP4.5 and RCP8.5 for the thermal expansion or for the internal variability effect (the difference in AR5 is about 8 cm, so it is maybe the latter?)

Yes, this is right. It is the thermosteric and dynamic SLR. The uncertainty of SLR reduction is also given here which is due to the internal variability. In the revision, we have removed the numbers due to the word limitation.

- Time period is not mentioned (later it turns out to be 2061-2080 relative to 1986-2005, this is quite crucial for comparisons)

In the revision, we have added the period (2061-2080). And since we only discuss the SLR reduction from RCP8.5 to RCP4.5, the base period (1986-2005) is not important. So we didn't mention it in the revised abstract.

Introduction

- It takes a very long time to get to the point, especially since the reader still has no idea where we're heading after the abstract

In our revision, we have tried to make our points clearer in the abstract. In the first paragraph, we also stated the purpose of this study at the end of the first paragraph rather than waiting until the end of the introduction to do so. So the revised manuscript should read better.

- L13-14 ‘the potential benefits of the sea level change’ sounds like ‘there are also positive sides to SLR’, while I suppose the message that is intended is: ‘can we get less sea level change if we would follow a lower emission scenario instead’?

This is a good comment. The reviewer is right. This is what we want to say. We have modified our abstract as “Here, by analyzing two sets of ensemble simulations from a climate model, we investigate potential SLR that may be avoided if a lower emission scenario is followed instead of business-as-usual one over the 21st Century and how it may be modulated by internal climate variability.”

- L30 ‘recent centuries’?

This has been removed

- L35 ‘mass component’ would be more correct. ‘eustatic’ is not really used anymore in this context as it’s actually defined as the sea-level change in a rigid ocean basin (check for instance Rovere et al, 2016, <http://link.springer.com/article/10.1007/s40641-016-0045-7>)

Thanks for this comment. The term is changed and the reference is added

- There is lots of repetition in L27-48; jumping back and forth between dynamics/ice, global/regional – could be shortened

This is a nice comment. In the revision, we have streamlined the discussion, and tried our best to reduce the redundancy. For example: “Global mean SLR is mostly determined by ocean mass changes, glacial isostatic adjustment, steric contribution, and groundwater mining and dam building. However, regional heterogeneous SLR is mostly related to the changes in Earth’s gravitational field and ocean dynamics. Both of these two processes do not change the global mean sea level, instead, they only redistribute the water mass, including heat and salt, within the ocean²². For example, as the currents and mass within the ocean shift, sea level rises in one area while falling in another, leading to an uneven change of local sea level^{16, 17, 28-31}. The mass loss from ice sheets reduces the gravitational force between the ice sheets and surrounding ocean water, and also induces a rebounding of the land^{2,31}, resulting in a higher than the global mean SLR away from melting ice sheets, but a lower than global mean SLR around the melting ice sheets.”

- L45 what is ‘melt-back’?

This term has been changed to “decrease” or “melt” throughout the paper, which will reduce the potential confusion.

- L49-52 repetition in sentence

This discussion has been removed in the revised manuscript.

- L54 moreover = however

Done.

- L58-L61: This is FINALLY explaining what will be investigated! Please state this in the abstract!

Very good comment. In our revision, we made this point clearer in our abstract and also moved this sentence to the first paragraph of the introduction section.

- L59 what is ‘global mean regional SLR’??

It should be “global mean and regional SLR”.

- L61-62 I would take out this sentence, it only confuses rather than clarify - unless the authors actually show a model simulation driven by RCP8.5 for the first half of the century and by RCP4.5 for the second half.

This sentence has been removed from our revision.

Methods

- L71-72 This is cryptic. Why is this done and what does it mean?
- L73-74 So there are 30 members in the historical run, then 30 go on for RCP8.5 and (the first?) 15 for RCP4.5? Why?
- L75 why 2080, it seems quite random? Using this period makes it also very hard to compare to IPCC (or other sources).

These three comments are very good. In our revision, we have tried to briefly mention why these experiments are done.

Yes, there are 30 ensemble members for the historical runs (1920-2005) and RCP8.5 (2006-2080), but only 15 members for RCP4.5 (2006-2080). More information can be found in the CESM large ensemble homepage (<http://www.cesm.ucar.edu/projects/community-projects/LENS/>). The original plan for this project was to investigate how the internal variability can affect the simulated and project climate in light of the CCSM3 large ensemble which was run from 2000-2061 under the A1B scenario. The run to 2080 is purely due to the limitation of the computational resources at the time the project was carried out. For

the same reason, the medium size ensemble for RCP4.5 is also restricted to the computational resources at that time.

For the CCSM3 large ensemble, we found significant regional climate differences due to the internal variability among different ensemble members (e.g., Deser et al., 2010, 2012a,b; Hu and Deser, 2013). These results motivate us to do a longer large ensemble simulation (1920-2080).

As indicated by Reviewer #1, the simulations stopped by 2080 make them hard to compare to IPCC simulations. Because of this reason, recently the RCP8.5 simulations have been extended to year 2100, but not for the RCP4.5 simulations. Thus here we can not extend our analysis beyond 2080.

*Deser, C., M. A. Alexander, S. -P. Xie, and A. S. Phillips (2010), Sea surface temperature variability: patterns and mechanisms, *Ann. Rev. Mar. Sci.*, **2010.2**, 115-143, doi:10.1146/annurev-marine-120408-151453.*

*Deser, C., A. S. Phillips, V. Bourdette, and H. Teng (2012a), Uncertainty in climate change projections: The role of internal variability, *Climate Dyn.*, **38**, 527-546, DOI 10.1007/s00382-010-0977-x.*

*Deser, C., R. Knutti, S. Solomon, and A. S. Phillips (2012b), Communication of the role of natural variability in future North American climate, *Nat. Clim. Change*, **2**, 775-779, doi: 10.1038/nclimate1562.*

*Hu, A. and C. Deser, 2013, Uncertainty in future regional sea level rise due to internal climate variability, *Geophys. Res. Lett.*, **40**, 2768-2772, doi:10.1002/grl50531*

- L77-79 this should be said in the introduction already

This sentence has removed from the revision.

- Is there no drift correction applied to the ocean variables?

Yes, there is a drift in model control run and the linear trend is removed from all data used here. In the revision, we have added a sentence to clarify this in the method section.

Global and Regional Mean SLR

- (should this subtitle be ‘global mean and regional SLR’?)

Yes, this subtitle is changed.

- L92: these numbers seem low compared to IPCC (or at least at the lower end), and the uncertainties (unclear if this is 1 sigma??) are very small, why?

Yes, the uncertainty is 1 sigma. For the global mean SLR due to thermal expansion, the CESM projection is similar to those in CMIP5, such as SLR in Figure 13.11 from the IPCC AR5 (red lines, thermal

expiation only) around 2070 is about 17.5 cm. As shown in Hu and Deser (2013) and Hu et al. (2017), the uncertainty for global mean SLR in an ensemble simulation is very small in comparison to regional SLR. One reason is all of these ensemble members are using the same model and the overall heat absorption by the ocean is similar under the same external forcing change. In the CMIP5 models, the uncertainty is most likely due to differences of the model physics and (horizontal and vertical) configurations. Differing climate model sensitivities will affect the efficiency of ocean heat uptake and where that heat is deposited within a particular model.

- L104: ‘in other words’. Suggest change ‘global warming’ to ‘greenhouse gas emissions’?

Done.

- L108: strange wording, how can SLR and the SLR trend be similar? (same confusing terms in the rest of this paragraph)

This sentence has been changed to “In general, both the pattern of the ensemble mean mid- and late-century regional SLR and the pattern of longterm (2006-2080) trends are similar between the two scenarios. Specifically, a SLR higher than the global mean is in the subtropical Pacific, South Atlantic, Arctic, part of the subpolar North Atlantic and part of Indian ocean, but a lower one in the Southern Ocean, subtropical North Atlantic, equatorial Pacific, southeast part of the South Pacific, and subpolar North Pacific in both scenarios. The similarity in pattern suggests that the underlying governing internal processes are similar for both scenarios and over these time scales, and that the ensemble mean SLR could be scaled by the strength of the greenhouse gas forcing³⁷.”

Dynamic Sea Level Mechanisms and Changes

- L164-165 this is true by definition, but only if the thermal expansion and dynamics are the only SLR terms considered. If ice mass changes and other contributions are added, even a place with below-average dynamics can be above the global mean total SLR.

The reviewer is right. This is potentially true. Since ice mass changes and other contributions are not included in our simulation, we cannot directly estimate this effect. A caveat is added at the end of paper to briefly discuss this.

- It would be good to include some discussion on the ability of the climate models to reproduce internal variability in the first place. How good is the magnitude, spatial distribution and timing of the variability in the model?

This is a nice suggestion. A brief discussion is mentioned in the revised paper on the simulated internal climate variability in comparison with the observed ones: “As shown in previous studies, these internal climate processes are simulated reasonably well in CESM1 in comparison to observations³⁸⁻⁴¹.”

- L198-200: The model shows a weakening AMOC and a positive NAO, seemingly in contradiction to this statement. Or does the statement in this sentence mean that the AMOC would have been even weaker if it weren't for the strengthening NAO?

The reviewer is right. Previous studies show that a positive NAO is related to a stronger AMOC since a positive NAO induces a colder than normal winter and a stronger deep convection in the subpolar North Atlantic. In our simulations, the AMOC weakens due to the warming of the upper ocean induced by the greenhouse gases, which strengthens the upper ocean stratification and weakens the deep convection and the AMOC. Thus, if the NAO did not have an upward trend, the AMOC could weaken further. That sentence has been changed to: "Earlier studies show that a longterm positive NAO strengthens the AMOC^{49,50}, suggesting the AMOC in RCP8.5 could have weakened more if there had been no upward NAO trend."

- L207 – There are quite some papers that have discussed this pattern in the Southern Ocean due to a shift in the ACC, it would be good to reference at least a couple here (for instance Böning, C. W., A. Dispert, M. Visbeck, S. R. Rintoul, and F. U. Schwarzkopf, 2008: The response of the Antarctic Circumpolar Current to recent climate change. *Nat. Geosci.*, 1, 864– 869, doi:[10.1038/ngeo362](https://doi.org/10.1038/ngeo362); Frankcombe, L., P. Spence, A. M. Hogg, M. H. England, and S. M. Griffies, 2013: Sea level changes forced by Southern Ocean winds. *Geophys. Res. Lett.*, 40, 5710–5715, doi:[10.1002/2013GL058104](https://doi.org/10.1002/2013GL058104).)

Thanks for this comment. These and other references have been added.

Summary

- Coming back to my comment reg L58-61, I thought that this was what would be investigated, but somehow by the end of the paper I'm not too sure anymore, because it is barely discussed in the summary.

In our revision, we have tried to make this clearer. Yes, this is still the focus of this paper. A new paragraph is added to discuss this better: "Our results further show an increase in variability of the PDO and STCs in the Pacific and the ACC in the Southern Ocean (especially in RCP8.5), which contributes to the overall larger SLR variability and less significant SLR reduction in these regions. The AMOC variability is reduced in the 21st century relative to 20th century and the variability is smaller in RCP8.5 than in RCP4.5, which contributes to less SLR variability in most parts of the Atlantic and a more significant SLR reduction, especially in the period of 2061-2080 due to a much larger weakening of the AMOC in RCP8.5. The variability of the NAO does not have long-term changes in our simulations for both scenarios, but its interannual variability does contribute to the across ensemble SLR variability."

- It is not that surprising that the differences between RCP4.5 and RCP8.5 are not very big on the short term, because the forcing only really starts to diverge later in the 21st century.

This is true, the equivalent CO₂ concentration between RCP4.5 and 8.5 is small at 2020, but increases to 20~30 ppmv by 2030, and to 50~60 ppmv by 2040. On average, we can assume a difference of ~30 ppmv for 2021-2040. The global mean SLR difference between these two scenarios averaged over 2021-2040 is 0.5 cm (a 10% reduction). However, although this small change in SLR is not surprising, it is still important to point this out, especially for people who are outside of our climate change field.

- It should maybe be mentioned somewhere early on that the ‘city values’ are actually a pretty big grid cell that is close to the location mentioned.

This is right. A couple of sentences are added in the method section to clarify this: “The SLR for selected global cities as shown in Supporting Figure 1 is defined as the SLR of the closest ocean grid point, which may minimize the actual SLR variability.”

Figures/tables:

- Figure 1: the trend period should be mentioned here

Actually, the trend period is mentioned in the original Figure 1 caption: “The decadal trend is the linear trend of the sea level rise from 2006 to 2080. The left panels are for RCP8.5 and right panels for RCP4.5.”

- Figure 2: I find it confusing that the time period leading to larger sea level rise is below the shorter period with less sea level rise

In the revision, we have added a black line to separate the top and bottom portions of the Figure 2 so it will be easier to read. We also added the global mean SLR lines in the plot to make the comparison with the global mean SLR easier.

- There is a typo in the caption of SupFig1. What are the uncertainties here, 1 sigma? What are the uncertainties anywhere in the paper?

In this figure, we did not give the uncertainty, instead we plot the global mean temperature and SLR for all individual members and the ensemble mean. In the paper, the uncertainties are represented by 1 sigma unless otherwise specified. The typo is corrected.

- SupTable1: caption should be “Global and regional sea level rise in 21st century (cm)”. Might it be better to put percentages of the global mean rather than cm?

The caption has changed. After considering this comment, we have added a new column for the percentages of SLR in selected cities relative to the global mean SLR.

- How come different places are below average between the two scenarios? (there are more in the RCP85 scenario?) -> it is exactly this that I would have expected to be discussed in this paper, does this have to do with reduced (or enhanced) internal variability?

The ensemble mean patterns of the SLR are associated with the ensemble mean trend of internal climate variability. For example, the higher than global mean SLR in the subtropical Pacific is related to the upward trend of the PDO and the strengthened STCs variability compounded with a strengthened ocean stratification, which enhances the variability of the subtropical Ekman convergence and downwelling, resulting in a higher SLR and SLR variability. This has been explained better in the revised manuscript.

- SupFig3: a line indicating the global mean would be handy. What is the grey dot at the global mean? Note that in the text this figure is mentioned in text before SupFig2.

A line representing the global mean has been added. We also removed the top panel of this figure. Now it only shows the SLR trend from 2006-2080. The order of the supporting figures has been rearranged.

“What is the grey dot at the global mean?” This is corrected.

- SupFig6: this should probably be done by a scatter plot: the line (and areas) suggests this data is continuous and connected. Also, I have no idea what this graph is supposed to tell me and how it supports the claims made in L132-136?

This figure and the related discussion have been removed from the revised manuscript.

Response to reviewer #2

General assessment

Unfortunately, I cannot recommend publication of the manuscript due to the reasons listed under “major comments”. For the paper to be published at all it would need significantly revised because there are a number of problems in the interpretation of the simulation results, but even after such revisions, I do not see it to rise to the standards in novelty and significance that is generally required within the NPG. However, in particular this question of significance should, of course, be decided by the editor.

Thanks for this general comment which have motivated us to do a lot more new analysis and focused more on the change of the across ensemble variability. We have revised our manuscript according to the suggestions made by reviewer #2.

Major comments

1. Confusing statements: The title promises a comparative investigation of internal variability and future trend with respect to sea level changes, but already in the abstract (and through-out the paper) the focus lies on the difference in global mean sea level change between different emission scenarios. This comparison has been done short of a million times and is not a high profile result. It took a while for me to understand what the authors mean by “internal variability”: They claim that most of the spatial differences of sea level changes arise from internal variability. I think that is by no means justified because the regional sea level changes will be in balance with the oceanic circulation and the wind stress changes and changes in both are likely to arise from changes in the climate system. The assumption that this is “internal variability” is particularly puzzling because the authors average them over a decade.

In our revision, we have tried to explain what we study here better to reduce potential confusion. The specific focus of this study is to determine what global and regional SLR could be avoided

when a moderate emission scenario (RCP4.5) is followed rather than a business as usual scenario (RCP8.5) and how internal climate processes will induce SLR uncertainty and contribute to regional SLR differences between the future scenarios. Our result basically show that due to the influence of the internal climate variability, the SLR reduction in RCP4.5 from RCP8.5 may not be statistically significant in certain regions even towards the end of the 21st century.

The reviewer is right that the regional sea level is in balance with the oceanic circulation and the wind stress. Both of these can change with changes of the external forcings. Since what we explore here is ensemble simulations using a single climate model and the identical external forcing for each set of the ensemble simulations, the forced changes of the ocean currents and winds represented as the ensemble mean is the same (or at least very similar) within each ensemble simulations. The differences between each of the ensemble members is caused mostly (if not totally) by the different time evolution of the internal climate processes, such as the PDO, and also by trends of those modes of variability (or changes in variance with time). As shown in our supporting Figure 15, the time evolution of the PDO is significantly different from one ensemble member to another, thus generating different responses of the wind and ocean currents, and different regional sea level change at any given time.

In different ensembles, the modulation of the internal variability by the different strength of the external forcing could be different. As shown in the new Figure 6, the external forcing does not insert significant modulation to the PDO or NAO, but does insert significant modulation to AMOC and STC (representing both buoyancy and wind driven circulations) relative to the 20th century. But these modulations between RCP8.5 and RCP4.5 in the 21st century are not statistically different from one to another. The internal variability (NAO, PDO, etc) affects both wind stress and ocean circulation, further affects the regional sea level rise, which is pointed out in the manuscript.

The definition of internal climate variability used here is not significantly different from many previous studies, such as Deser et al., 2010, 2012a, b. This variability exists in the nature world and does not necessarily depend on the external forcing for its existence.

As shown in this manuscript, SLR variability induced by internal processes over a 20-year mean is still very large. If we were to analyze the SLR variability induced by interannual variability, the SLR variability could become even larger. Moreover, it would be really hard to figure out what processes play a more dominant role on SLR variability at interannual time scale. Therefore, we choose to use 20-year mean. On the other hand, averaging over a decade can show the tendency toward a positive or negative phase of one of these modes. In fact, for the decadal prediction studies, researchers focus on a 5-year mean as a target, not the interannual variability, which is also due to more complicated processes governing this interannual variability.

2. Missing novelty: The thermal expansion of the ocean has been reported a number of times. Previous studies, for example by the first author, even used ensembles of models. The comparison between different warming scenarios is a standard result and I do not see how the paper provides anything new or special with respect to method, approach or result.

It is true that SLR due to thermal expansion of seawater has been reported by various previous works. What is new here is by using a set of unique large ensembles, we can explore how internal climate variability (which previously has been considered as noise relative to the anthropogenic forcing induced climate changes) can modulate the regional sea level, and specifically how that modulation might be different when by following a lower emission scenario instead of a higher one since the point of the paper is to determine what SLR could be avoided if following RCP4.5 rather than RCP8.5. This has not been systematically explored in previous studies, including Hu and Deser (2013). Our results indeed show that statistically significant SLR between future scenarios may not show up in certain regions even towards the end of the 21st century. This is important, especially for the policy makers. A quick fix for the global mean temperature can be achieved in the short term by reducing the greenhouse gas emission. Mitigating SLR may not be achievable in the short term due to the influence of the internal climate variability and also because the SLR is an integrated property of the climate system which will respond to any changes in external forcing in a much longer time scale.

3. Overselling: a. The thermal expansion constitutes only 40% of the global mean sea level rise. The title, abstract and most of the paper are written as if it was equal to the global mean sea level rise. That is not appropriate.

It is true that the seawater thermal expansion constitutes only 40% of the recent observed global mean sea level rise. It was higher in the first about 80 years of the 20th century. In the future, the projected contribution from ice sheets and glaciers will become much more important. This will not reduce the SLR uncertainty due to internal climate processes. Thus, it is still important to study how these internal climate variability will modulate the regional sea level change.

On the other hand, the current generation of models do not include ice sheet melting and the associated gravitational forcing change, isostatic adjustments, and land rebound. It's possible that these effects will not be included in climate models in the next 10 years. Simple offline calculations of these effects and linearly adding these effects onto the model produced dynamic+thermal steric sea level field has been done, such as IPCC AR5 and a few newer studies. However, there is no systematic study to test whether these effects can be linearly added up, how the ocean dynamics will respond to these effects is still unclear and a research question. Thus, the SLR uncertainty sampled here may be the lower bound of the real SLR uncertainty. But this still provide useful information for the policy maker.

In the abstract, due to the word limitation, we cannot include too many details, but in our manuscript we have clearly stated that we only study the dynamic and thermosteric sea level change. In the revision, we have made this point more clear in both the method section and the summary section. In fact, we include this sentence in the methods section: "This limits our ability, allowing only estimation of thermosteric and dynamic SLR, which together accounts for approximately 40% of the observed global mean SLR for recent decades^{2,29} and may become even less by the end of this century²." When we say "global mean," we are referring to the global mean of the thermosteric and dynamic SLR, the parts we are able to capture with the model.

Nevertheless, we have made our points more clear and pointed out our focus is on thermosteric and dynamic sea level rise in three places: abstract, the first paragraph of the introduction section, and the first paragraph of the method section, in order to reduce any potential confusion.

b. The paper falls behind previous studies used an ensemble of different climate models for similar analysis by the same lead author. The model differences are significant and it is not clear why the reader is to believe that the results from one model is providing the full answer. It is however presented as the full answer.

The reviewer is right that the results shown here might be model dependent. The major focus of this study is the internal variability for which large ensembles are needed, and the CESM project provided large ensembles of 20th Century, RCP8.5 and RCP4.5. Internal variability could be over- or under-represented in our model; however, this will only change our results presented here quantitatively, not qualitatively. The results shown here still providing useful information. For a multi-model ensemble, the uncertainty sampled is related to both the internal climate variability and the differences in model configuration. The latter may be more important and represent our imperfect knowledge on the natural world. At the end of the paper, we added a short discussion on this.

4. The literature is not up-to-date. Although there are some references of 2016, most of the literature reflects the scientific state of five years ago. Practically no reference to the work after the latest assessment report of the IPCC and almost no reference to glacier or ice sheet contributions.

Thanks for this comment. We have surveyed new literature and updated our reference list to reflect the advancement of our understanding on regional SLR in recent years.

5. To provide specific sea level rise values for different cities is problematic if only the thermal expansion of a coarse resolution climate model is provided. The spatial precision that is suggested by selection of a city is not appropriate when the numbers provided (1) are from a coarse resolution model, (2) only cover 40% of the currently observed sea level rise and (3) do not include potential contributions for example from tectonic uplift which can be as strong as the regional sea level or the global sea level rise.

This is a reasonable comment. However, as we have stated in our manuscript, the SLR discussed here includes both SLR due to thermal expansion of seawater and the dynamic sea level. If only the thermosteric sea level is included, it does not have a regional pattern because it was added onto the dynamic sea level as a global mean number. Regional SLR differences are due only to the dynamic effect (the changes of the ocean currents and dynamics due to changes in wind and buoyancy forcings) and can add to or subtract from the global mean regardless of what the definition of global mean is; therefore, the patterns of regional sea level under different future scenarios are important for regional planning in coastal areas This is clearly stated in our original manuscript. Moreover, the accuracy of the SLR for specific cities depends on how well the model can simulate the major oceanic currents. As shown in the CESM1 special issue and other previous documentations of CCSM model series, the CESM1 is capable of simulating the

major ocean currents well in this one degree version of the model. Biases do exist and in some regions, these biases can be more serious than other regions. Nevertheless, CESM is one of the best state-of-art climate models in the world. We do think the existing model biases would affect our results quantitatively, but not qualitatively. The results discussed here can still provide reasonable information for the policy makers.

Additionally, the SLR noted for specific cities are actually part of a larger, regional response of SLR because it is driven by large scale climate variability (like the NAO, PDO, etc.); therefore, we could make generalizations about specific cities based on these larger patterns even if we had not done the analysis at specific grid points. For example, for the eastern seaboard of the U.S., studies have shown that the northern part will experience a lowering of sea level while the southern portion will experience a rising sea level relative to the global mean. The cities we have chosen in these regions match that pattern.

As we have mentioned in an earlier response, CESM does not have an active ice sheet model and we are not capable of assessing the uncertainties of the SLR due to the ice sheet response to the external forcing change, and it does not include the tectonic uplift either. The reviewer is right that due to these model deficiencies, cautions need to be made in interpretation these model results. We have briefly discussed these in our revised manuscript.

6. The results of the simulation are generally just reported and not explained (for example by ocean circulation changes).

In general, we have two major parts in our paper. One is to show the sea level rise on global and regional scales and the uncertainty of the sea level rise; the other part is the explanation of how the internal variability affects the sea level change in terms of the dynamic sea level change. In our revision, we have tried to do a better job of explaining how the sea level rise patterns are related to the internal variability trends, and how the internal variability affects the sea level rise uncertainty.

Selected minor comments

1. Literature: The literature seems outdated and at places very imprecise. Some example: To cite the IPCC in line 37 for the thermosteric and halosteric effect is like citing a text book on the Navier-Stokes equation. There is no use to it in a scientific publication. Also the collection of references, for example in lines 58 and 183, is not very helpful and seems to merely serve the purpose of putting a lot of papers in the reference list as opposed to inform the reader about results previously reported in the literature.

Thanks for this comment. We have tried to survey the more recent literature in our revised manuscript. The sentences of the original lines 58 and 183 have been modified. The references are reorganized. On the other hand, although the IPCC report is about 5 years old, it still provides valuable knowledge to us.

2. Fig. 2: The colouring of the dots should be explained in the figure caption.

Explanation of the colored dots is added.

Response to Reviewer #3

Overview

This paper represents a nice overview of various papers and IPCC reports. Much of the “results” have been published in previous papers: the global projections are in the IPCC and the regional projections are also found in the IPCC and various sea level rise intercomparison papers. This includes the impact of internal variability, although I acknowledge the discussion found in the intercomparison papers is not as clear/direct as in this paper. The new part is a clear discussion of the impact of mitigation on the regional projections and using many ensemble members of 1 model. BY using the many ensemble member of 1 model the influence of variability is reliably assessed. The papers previous assessed the impact of variability by using a multi-model ensemble although this mixes response differences and variability. This paper represents a clear improvement of the discussion of SLR. My recommendation is to publish the paper since it is very readable and understandable to most non-specialists. Further it presents some new and interesting results. That said, I have a number of specific comments the authors should address before the paper is published. This are all relatively easy to address. I would not have to see the paper again before publication.

We appreciate reviewer #3's constructive and insightful comments. The manuscript has been revised according to these comments. A detailed point-by-point response is listed below.

Specific Comments

1. Line 1 – Title – I found the title confusing and/or misleading. A suggestion to change it – Influence of mitigation and internal climate variability on the projections of future regional sea level rise.

Now the title changed to “Internal climate variability and potentially avoided impacts of the projected future regional sea level rise”

2. Lines 9 – global mean sea level rise is also rising – reads funny. Change to global mean sea level rise also rises.

Done

3. line 17 – RCP – I would argue that “RCP scenario” is jargon. Change to “emission scenario”.

Done

4. Line 19 – marginal – Is “small” better?

Done.

5. Line 20 – Delete “very large”. Change to “larger”. 10cm (the difference between sites) of SLR may or may not be large. Also, this assessment of “large” depends on the time scale in view.

Done.

6. Line 21 – Shouldn't the Yin et al. (ref 20) be cited here?

Cited.

7. Line 26 – by about 1 – Change “by” to “of”.

Done

8. line 27 – Change “this sea level rise” to “the observed sea level rise”. It makes the meaning clearer.

Done

9. Line 42 Change “total heat” global heat”. “Total” is the wrong word here.

Done.

10. Lines 43 -45 – Could add reference to Gregory et al. model intercomparison SLR paper to list.

Done.

2001: Comparison of results from several AOGCMs for global and regional sea-level change 1900-2100. *Climate Dynamics*, 18(3/4), 225-240.

11. Line 99 – Somewhere near here, it should be noted that the values being discussed are strongly dependent on the time scale of interest. Also, add “by 2080” after “29%”.

Done.

12. Line 105 – Could add a reference to Stouffer et al.
1999: Response of a coupled ocean-atmosphere model to increasing atmospheric carbon dioxide: Sensitivity to the rate of increase. *Journal of Climate*, 12(8), 2224-2237.

Done.

13. Line 113 – After “for these scenarios” add “and over these time scales”.

Done

14. Line 178 – Could cite any one of a number of old (circa 1990's) papers which highlight these processes. This is not a new result.

New reference is added

15. Lines 210 – 221 – Should not there be an eddy caveat here?

A sentence is added to state the potential influence of eddy “As ACC is an eddy rich region, inclusion of realistically simulated eddy effect may modulate our results slightly as suggested by previous studies^{52,54}.”

16. Lines 222- 225 – Again, this is an old result. Please cite the early papers.

Two new references are added:

Boning, C. W., A. Dispert, M. Visbeck, S. R. Rintoul, and F. U. Schwarzkopf, The response of the Antarctic circumpolar current to recent climate change. Nature Geosciences, 1, 864-869, doi:10.1038/ngeo362 (2008).

Bi, D., W. F. Budd, A. C. Hirst, and X. Wu, Response of the antarctic circumpolar current transport to global warming in a coupled model. GRL, 29, 2173, doi: 10.1029/2002GL015919 (2002).

17. Line 239 – Compared to other CMIP models or just the ensemble of this model? Not clear.

This is only for CESM1. The sentence has been modified to clarify this.

18. Lines 273 – 281 – Several caveats are missing here.

These runs are missing

1. land water storage and adding (dams and deep water pumping).
2. land ice melt/freezing.
3. gravity changes

Therefore, the details will change when model can account for those changes.

Caveats have been added in the discussion section. This is a very nice suggestion.

19. Figure 1 caption – Line 432 – “top left” should be “top right”.

All panels are weighted by those global mean values – what does this mean? The global mean is removed?

Line 432 is corrected. Here all values are divided by the global mean, it is not a removal of the global mean. The caption has changed.

20. Figure 2 caption - - line 443 – variability – Maximum to minimum? ... or what?

It is one standard deviation to represent the variability of the SLR. The caption has changed: “Figure 2. 20-year mean sea level rise for selected cities. This mean sea level rise is relative to the mean of 1986-2005. The upper part is for the mean of 2021-2040 and the lower one for 2061-2080. The geographic location of these cities is given in Supplementary Figure 3. The solid dots are the ensemble mean sea level rise for RCP8.5 and open circles the ensemble mean sea level rise for RCP4.5. The bars indicate ensemble variability (± 1 standard deviation). The unit is cm. The brown and light blue line represent the ensemble global mean SLR for RCP8.5 and

RCP4.5, respectively. The color coded dots/open circles represent east (green) and west (brown) Pacific coasts, west (black) and east (red) Atlantic coasts, and Indian Ocean coast (blue).”

21. Figure 3 – What color scale goes with what figure? I could not figure it out. Sorry for the pun. Ratios are displayed according to the caption. I do not see a color to fit this. Are the values percentage change?

The left color bar is for panels a and b, and the right color bar is for the panels c and d. The top color bar labels are for panel c and the bottom color bar labels are for panel d. This caption has been changed to make it clearer.

22. Figure 4 caption – line 457 – I assume this is a ensemble mean. Correct?

*Yes, it is correct. Actually the caption also indicates this **ensemble mean** in our original manuscript: “The top panel is the ensemble mean sea level pressure (hPa) and surface wind (m/s) for the late 20th century (20C, averaged over 1986-2005). The mid and bottom panels are the **ensemble mean** decadal trend of the sea level pressure (hPa/decade) and wind (m/s/decade) from 2006-2080 for RCP8.5 and 4.5.”*

Supplement

23. Table 1 and 2 – Column labels – Use one style – All caps or not. There is a mixture currently.

Done

24. Figure 1 – Very hard to see individual model lines.

This is right. This plot is not intended to let readers read individual lines since there are 30 individual lines there. This plot just gave the reader a sense that the spread of the global mean surface temperature and sea level rise is small.

25. Figure 4 – Really hard to see anything meaningful. Chose a few panels to highlight.

Good suggestion. We have debated a lot on how to present this figure. We did some selected cities and then changed to all cities. In the revision, we choose six most representative cities to show the various response of the regional sea level to these two RCP scenarios.

26. Figure 5 caption/figure – What is SLC? Define.

It is SLR. It has been corrected.

27. Figure 6 – Green line is very hard to see.

Supporting figure 6 is removed from the revised manuscript.

28. Figure 12 – a) and b) is figure not defined in caption.

In the revision, we have defined panels a and b. a is the Drake Passage transport for the CESM1 large ensemble (RCP8.5) and medium ensemble (RCP4.5), panel b is the same as a but for CMIP5 simulations of CESM1.

29. Figure 14 – I think the 20C panel should be on top. The RCP8.5 panel should be the third down from the top.

To keep it consistent, we have put 20C on top, but RCP8.5 in the middle since we always mention RCP8.5 before RCP4.5.

Reviewer #1 (Remarks to the Author):

The authors have done quite some work and the paper is improved wrt the previous version: there is a larger focus on the topic of internal variability. However, I still find some things not clear, particularly in the first part of the paper – comments are provided below.

As was already mentioned by one of the previous reviewers, I think still the title is confusing/misleading: it doesn't properly describe the content of the paper.

- 'potentially avoided impacts' – this paper is not about impacts, it is about sea-level change projections.
- 'potentially avoided impacts (...) of (...) sea level rise' does not make sense.
- I think the title should focus on the effect of internal variability (which is the interesting and novel bit), not on the rcp4.5 vs rcp8.5 change (as the latter is not exactly novel)

Although it is now clearer that the paper only assesses thermosteric and dynamic SLR, but there is still some room for improvement. In L31, I think it should read 'could modulate regional SLR resulting from steric and dynamic processes, and show how much SLR could be avoided ...'

In l341 'the ensemble mean steric and dynamic SLR'

btw, 'thermosteric and dynamic' should be 'steric and dynamic' throughout (unless when discussing the global mean), because there are also halosteric effects that play a role in the regional SLR

L 44: global mean SLR is not determined by GIA (GIA is more important for regional change). For global mean it is not steric, but thermosteric that matters (halosteric averages out in the global mean). I would think that groundwater mining and dam building fall under the mass component?

L58: "However, the uncertainties from the internal climate processes will not decrease, especially on decadal timescales." Might be good to explain why not?

L73 "may become even less by the end of this century" the percentage may become less, but the absolute amount will probably increase

L89 suggest change 'minimize' to 'underestimate'. Also: explain why this is the case.

L97-98: I don't quite see the relevance of giving a century-average decadal trend – not particularly different message from century-averaged yearly rates really. Wouldn't it be more relevant to give the (decadal) trend in 2006 and in 2080 rather than the century average – given the projected acceleration in SLR?

L106-7: 'as reported before (references to IPCC and other SLR projections papers would be in order here)'

Is the message of the paper that internal variability does not change under climate change? Or is this an assumption? (e.g. l113-114)

L129/suppFig6: are these ratios of rates, or of cumulative change? How is the different ensemble size taken into account?

L138-139: this is also shown in e.g. Little et al <http://journals.ametsoc.org/doi/full/10.1175/JCLI-D-14-00453.1>. Might be good to acknowledge that there are other papers that have looked at the uncertainties as a result of internal variability

L191: would 'Drivers of Ocean Variability' or something along those lines perhaps be a better title

for this section?

L347-352; In the summary, the balance between novel findings (internal variability) and less novel findings (differences between RCPs) could be more prominent? Now the internal variability results almost feel as some sort of afterthought?

L372: but the impacts of coupling are small as shown in several studies (Agarwal et al. 2015; Howard et al. 2014; Slangen and Lenaerts 2016)

References

- Agarwal, N., J. H. Jungclaus, A. Köhl, C. R. Mechoso, and D. Stammer, 2015: Additional contributions to CMIP5 regional sea level projections resulting from Greenland and Antarctic ice mass loss. *Environ. Res. Lett.*, 10, 74008, doi:10.1088/1748-9326/10/7/074008.
- Howard, T., and Coauthors, 2014: The land-ice contribution to 21st-century dynamic sea level rise. *Ocean Sci.*, 10, 485–500, doi:10.5194/os-10-485-2014.
- Slangen, A. B. A., and J. T. M. Lenaerts, 2016: The sea level response to ice sheet freshwater forcing in the Community Earth System Model. *Environ. Res. Lett.*, 11, 104002, doi:10.1088/1748-9326/11/10/104002.

Additional review points at the Editors request

My responses are in red, authors comments in blue, original Rev#2 comments in black, underlined the Editors questions.

Confounding of thermal expansion and global mean sea-level rise (point 3)

3. Overselling: a. The thermal expansion constitutes only 40% of the global mean sea level rise. The title, abstract and most of the paper are written as if it was equal to the global mean sea level rise. That is not appropriate.

I agree with the reviewer that the original manuscript was 'overselling' steric/dynamic change as SLR, and I made a similar comment in my first review. The authors have made some changes to this effect, but as I mentioned in my own re-review: I think there is still room for improvement. Probably this should already be done in the title, and early in the abstract (it is done now but still a bit implicit): it has to be absolutely clear. E.g. in l11: '... we investigate potential SLR **as a result of steric and dynamic oceanographic effects alone** that may be avoided...'. I think it is better to 'oversignal' than to 'undersignal' this – just to make sure that confusion is avoided.

In the response to the reviewer, the authors note: "*In the future, the projected contribution from ice sheets and glaciers will become much more important. This will not reduce the SLR uncertainty due to internal climate processes. Thus, it is still important to study how these internal climate variability will modulate the regional sea level change.*" Perhaps a sentence to this effect should be in the text a bit more explicitly, around lines 58-59?

The authors also note "*Thus, the SLR uncertainty sampled here may be the lower bound of the real SLR uncertainty.*" Indeed, this is very likely: given that this is 'only' 1 model, and 'only' 1 component of SLR. This may be worth expanding on in the text.

b. The paper falls behind previous studies used an ensemble of different climate models for similar analysis by the same lead author. The model differences are significant and it is not clear why the reader is to believe that the results from one model is providing the full answer. It is however presented as the full answer.

"(...) At the end of the paper, we added a short discussion on this. "

Indeed I see in the paper a small discussion on the internal variability in the CESM model (lines 353-363). However, I do not really see the answer to the reviewers' question, namely: what is the effect of the use of 'only' 1 model while in previous work multiple models were analysed? I know this is difficult, but I agree with the reviewer it needs to be addressed.

discussion of model deficiencies (point 5)

5. To provide specific sea level rise values for different cities is problematic if only the thermal expansion of a coarse resolution climate model is provided. The spatial precision that is suggested by selection of a city is not appropriate when the numbers provided (1) are from a coarse resolution model, (2) only cover 40% of the currently observed sea level rise and (3) do not include potential contributions for example from tectonic uplift which can be as strong as the regional sea level or the global sea level rise.

This is a reasonable comment. However, as we have stated in our manuscript, the SLR discussed here includes both SLR due to thermal expansion of seawater and the dynamic sea level. If only the thermosteric sea level is included, it does not have a regional pattern because it was added onto the dynamic sea level as a global mean number. Regional SLR differences are due only to the dynamic effect (the changes of the ocean currents and dynamics due to changes in wind and buoyancy forcings) and can add to or subtract from the global mean regardless of what the definition of global mean is; therefore, the patterns of regional sea level under different future scenarios are important for regional planning in coastal areas This is clearly stated in our original manuscript.

I agree with the reviewer that using 'city' values can be problematic. The authors have in this new manuscript added a clarification to the methods section that the city values are taken from the nearest grid point. However, as I mentioned in my re-review, the sentence 'which may minimize the actual SLR variability' is a bit awkward and I would suggest to change it.

In the light of Rev#2's comments, I would strongly suggest to put some more explanation on why there could be differences between the 'open ocean' and the actual coastal change. E.g. missing processes, missing topography, propagation of steric signals to the coast. It deserves a bit more discussion than the current half sentence, as there is a 'danger' that people will take these values literally!

Additionally, the SLR noted for specific cities are actually part of a larger, regional response of SLR because it is driven by large scale climate variability (like the NAO, PDO, etc.); therefore, we could make generalizations about specific cities based on these larger patterns even if we had not done the analysis at specific grid points.

I would suggest to actually put this in the text, that would be very helpful.

the level of the discussion of results (point 6)

6. The results of the simulation are generally just reported and not explained (for example by ocean circulation changes).

In general, we have two major parts in our paper. One is to show the sea level rise on global and regional scales and the uncertainty of the sea level rise; the other part is the explanation of how the internal variability affects the sea level change in terms of the dynamic sea level change. In our revision, we have tried to do a better job of explaining how the sea level rise patterns are related to the internal variability trends, and how the internal variability affects the sea level rise uncertainty.

I agree with Rev#2 that this was the case in the original manuscript. I think that in the new version the focus is much more in the internal climate processes and how they affect dynamic sea level, which is the new & noteworthy bit of the paper.

Response to Reviewer #1

We thank reviewer #1 for his insight and constructive comments. We have revised our manuscript based on these comments. Here our response is in blue color bold Comic Sans MS font.

As was already mentioned by one of the previous reviewers, I think still the title is confusing/misleading: it doesn't properly describe the content of the paper.

- 'potentially avoided impacts' – this paper is not about impacts, it is about sea-level change projections.
- 'potentially avoided impacts (...) of (...) sea level rise' does not make sense.
- I think the title should focus on the effect of internal variability (which is the interesting and novel bit), not on the rcp4.5 vs rcp8.5 change (as the latter is not exactly novel)

Now the title has been changed to "Internal climate variability and projected future regional steric and dynamic sea level rise"

Although it is now clearer that the paper only assesses thermosteric and dynamic SLR, but there is still some room for improvement. In L31, I think it should read 'could modulate regional SLR resulting from steric and dynamic processes, and show how much SLR could be avoided ...'

This has been changed.

In l341 'the ensemble mean steric and dynamic SLR'

btw, 'thermosteric and dynamic' should be 'steric and dynamic' throughout (unless when discussing the global mean), because there are also halosteric effects that play a role in the regional SLR

Thanks for this comment. We have kept this consistent.

L 44: global mean SLR is not determined by GIA (GIA is more important for regional change). For global mean it is not steric, but thermosteric that matters (halosteric averages out in the global mean). I would think that groundwater mining and dam building fall under the mass component?

The authors agree with the reviewer that GIA does not significantly affect the global mean SLR. This sentence has changed to "Global mean SLR is mostly determined by ocean mass changes, and the steric contribution."

L58: "However, the uncertainties from the internal climate processes will not decrease, especially on decadal timescales." Might be good to explain why not?

Good suggestion. This sentence has changed to: "Although the projected contribution from ice sheets and glaciers on SLR in the future will become much more important, whether the SLR uncertainties from the internal climate processes will change is still unknown, especially on decadal timescales. Since the existence of these internal climate processes is not model dependent and they are physical processes naturally occurring in the climate system, it is still extremely important to study how internal climate variability will modulate the projected regional SLR from steric and dynamic contributions."

L73 "may become even less by the end of this century" the percentage may become less, but the absolute amount will probably increase

This is right. The sentence has been changed to: "This limits our ability, allowing only estimation of thermosteric and dynamic SLR, which together accounts for approximately 40% of the observed global mean SLR for recent decades^{2,32} and may become even less percentagewise to the total SLR by the end of this century (although the absolute contribution from steric and dynamic SLR increases)^{2,19,20}."

L89 suggest change 'minimize' to 'underestimate'. Also: explain why this is the case.

This has changed to: "To link the potential impact of the SLR to human societies, here we use the SLR for selected global cities as shown in Supplementary Figure 1 as examples. The SLR for these cities is defined as the SLR of the closest ocean grid point, which may underestimate the actual SLR variability due the unresolved physical processes, such as ocean eddies and the detailed shape of coastlines. Additionally, the SLR noted for specific cities in this work is actually part of a larger, regional response of SLR to changes in the external forcing because the SLR is driven by large scale climate variability (like the NAO, PDO, etc.); therefore, we can make generalizations about specific cities based on these larger patterns even if we had not done the analysis at specific grid points. In reality, the coastal SLR is not only controlled by large scale climate patterns, but also affected by local winds on a spatial scale of tens kilometers in association with the ocean topography and shape of the coastlines (may also be affected by changes of tides in different timescales). If these processes were included in our simulations, it certainly would induce larger SLR variability in these coastal cities.

In fact, the response of the sub-grid-scale ocean to severe weather can also affect the sea level changes.

L97-98: I don't quite see the relevance of giving a century-average decadal trend – not particularly different message from century-averaged yearly rates really. Wouldn't it be more relevant to give the (decadal) trend in 2006 and in 2080 rather than the century average – given the projected acceleration in SLR?

Good suggestion. We have added a paragraph to discuss the rate of SLR changes in the late 20th century, the near future (2021-2040) and towards the end of the 21st century (2061-2080). "On the other hand, the rate of the GMST and global mean steric SLR changes is more significant than their respective mean changes as indicated by a previous study³⁷. The mean rate of GMST change in the late 20th century in CESM1 ensemble (1986-2005) is $0.19 \pm 0.14^\circ\text{C}/\text{decade}$, increasing to $0.39 \pm 0.14^\circ\text{C}/\text{decade}$ for RCP8.5 and $0.26 \pm 0.13^\circ\text{C}/\text{decade}$ for RCP4.5 in 2021-2040, and to $0.54 \pm 0.14^\circ\text{C}/\text{decade}$ for RCP8.5 and $0.22 \pm 0.12^\circ\text{C}/\text{decade}$ for RCP4.5 in 2061-2080, suggesting that the rate of GMST changes slows down later in the century for the lower emission scenario with continuous increase for the high emission scenario (Supplementary Figure 2c). Similarly, the rate of the global mean steric SLR increases from $0.63 \pm 0.18 \text{ cm}/\text{decade}$ during 1986-2005 to $2.16 \pm 0.19 \text{ cm}/\text{decade}$ for RCP8.5 and $1.76 \pm 0.18 \text{ cm}/\text{decade}$ in 2021-2040, and to $3.95 \pm 0.18 \text{ cm}/\text{decade}$ for RCP8.5 and $2.33 \pm 0.16 \text{ cm}/\text{decade}$ in 2061-2080, respectively (Supplementary Figure 2d). For RCP8.5, with unabated GMST increasing rate, the rate of global mean steric SLR is nearly doubled in 2061-2080 relative to that in 2021-2040. Although the rate of GMST changes decreases in RCP4.5 towards the end of 21st century, the rate of global steric SLR change increases continuously, which reinforces the points that reducing the greenhouse gas emission would not result in an immediate reduction in global mean SLR^{35,36}."

L106-7: 'as reported before (references to IPCC and other SLR projections papers would be in order here)'

References are added.

Is the message of the paper that internal variability does not change under climate change? Or is this an assumption? (e.g. 1113-114)

No. Here, we specifically describe the changes of the ensemble mean SLR. These ensemble mean SLRs are not significantly affected by the internal climate processes as long as the ensemble size is large enough. Otherwise, with a small ensemble size, the regional ensemble mean SLR can still be affected by the internal climate variability. When we discuss the change of internal variability, such as AMOC, PDO, NAO etc., one can clearly see some of the internal variability changes more (e.g., AMOC) than other internal variability (e.g., NAO). Thus we do not assume that the internal climate variability would not change under changing climate.

L129/suppFig6: are these ratios of rates, or of cumulative change? How is the different ensemble size taken into account?

We did not define this clearly enough. Now a new sentence is added in the supporting Figure 6 as "The intra-ensemble SLR variance is calculated as the variance across each individual ensembles." In fact, we calculate the decadal mean first for each ensemble members and then calculate the variance within this ensemble.

Here we did not take into account the potential influence of the different ensemble size. Our previous analyses show that an ensemble size of 15 is good enough for most regional SLRs, increasing the ensemble members would not affect the ensemble mean SLR much, but this increase in ensemble size does affect the intra-ensemble variance.

L138-139: this is also shown in e.g. Little et al <http://journals.ametsoc.org/doi/full/10.1175/JCLI-D-14-00453.1>. Might be good to acknowledge that there are other papers that have looked at the uncertainties as a result of internal variability

Reference is added. Thanks.

L191: would 'Drivers of Ocean Variability' or something along those lines perhaps be a better title for this section?

Agree. It has been changed to "Drivers of Ocean Variability and Change"

L347-352; In the summary, the balance between novel findings (internal variability) and less

novel findings (differences between RCPs) could be more prominent? Now the internal variability results almost feel as some sort of afterthought?

Thanks for this comment. For this part, possibly we didn't make our point clear enough. The sentence has changed to "Moreover, due to the non-uniform pattern of regional SLR associated with ocean dynamics, the ensemble mean SLR reduction may not be statistically significant even in the long-term in regions such as Australia and the Philippines due to enhanced internal climate variability, but can be very significant in other regions such as the east coast of North America which exhibit lessened internal climate variability."

L372: but the impacts of coupling are small as shown in several studies (Agarwal et al. 2015; Howard et al. 2014; Slangen and Lenaerts 2016)

Thanks for these references. However, in all these references and other studies of similar type, the model used cannot explicitly simulate the gravitational effect and the runoff water either distributed as virtual salt flux or as freshwater flux into the ocean without the gravitational effect. This gravitation effect is added offline which does not show whether the coupling of this gravitational effect with ocean dynamics will change the DSL. So in our opinion, this is still a research question and worth investigation more in the future when the coupled GCMs are capable of simulating the gravitational effect online.

References

Agarwal, N., J. H. Jungclaus, A. Köhl, C. R. Mechoso, and D. Stammer, 2015: Additional contributions to CMIP5 regional sea level projections resulting from Greenland and Antarctic ice mass loss. *Environ. Res. Lett.*, 10, 74008, doi:10.1088/1748-9326/10/7/074008.
Howard, T., and Coauthors, 2014: The land-ice contribution to 21st-century dynamic sea level rise. *Ocean Sci.*, 10, 485–500, doi:10.5194/os-10-485-2014.
Slangen, A. B. A., and J. T. M. Lenaerts, 2016: The sea level response to ice sheet freshwater forcing in the Community Earth System Model. *Environ. Res. Lett.*, 11, 104002, doi:10.1088/1748-9326/11/10/104002.

Response to comments from Review #1's evaluation on Review #2's comments and our previous response:

We thank Reviewer #1 for taking additional effort to evaluate our manuscript and thank for his/her insightful and constructive comments. We have further revised our manuscript based on these comments.

Additional review points at the Editors request

My responses are in red, authors comments in blue, original Rev#2 comments in black, underlined the Editors questions.

Confounding of thermal expansion and global mean sea-level rise (point 3)

3. Overselling: a. The thermal expansion constitutes only 40% of the global mean sea level rise. The title, abstract and most of the paper are written as if it was equal to the global mean sea level rise. That is not appropriate.

I agree with the reviewer that the original manuscript was ‘overselling’ steric/dynamic change as SLR, and I made a similar comment in my first review. The authors have made some changes to this effect, but as I mentioned in my own re-review: I think there is still room for improvement. Probably this should already be done in the title, and early in the abstract (it is done now but still a bit implicit): it has to be absolutely clear. E.g. in 111: ‘... we investigate potential SLR **as a result of steric and dynamic oceanographic effects alone** that may be avoided...’. I think it is better to ‘oversignal’ than to ‘undersignal’ this – just to make sure that confusion is avoided. In the response to the reviewer, the authors note: “*In the future, the projected contribution from ice sheets and glaciers will become much more important. This will not reduce the SLR uncertainty due to internal climate processes. Thus, it is still important to study how these internal climate variability will modulate the regional sea level change.*” Perhaps a sentence to this effect should be in the text a bit more explicitly, around lines 58-59?

The authors also note “*Thus, the SLR uncertainty sampled here may be the lower bound of the real SLR uncertainty.*” Indeed, this is very likely: given that this is ‘only’ 1 model, and ‘only’ 1 component of SLR. This may be worth expanding on in the text.

We appreciate these comments. We have revised our manuscript accordingly. In the introduction section, we have edited the text as “Although the projected contribution from ice sheets and glaciers on SLR in the future will become much more important, whether the SLR uncertainties from the internal climate processes will change is still unknown, especially on decadal timescales. Since the existence of these internal climate processes is not model dependent and they are physical processes naturally occurring in the climate system, it is still extremely important to study how internal climate variability will modulate the projected regional SLR from steric and dynamic contributions.” (lines 60-66)

In the discussion section, we added “Moreover, since only the steric and dynamic SLR has been examined here, the SLR uncertainty sampled here may be the lower bound of the real SLR uncertainty. Lastly, if multi-models are used, the differences in model physics and the horizontal-vertical configuration can certainly widen the uncertainty sampled here.”

b. The paper falls behind previous studies used an ensemble of different climate models for similar analysis by the same lead author. The model differences are significant and it is not clear

why the reader is to believe that the results from one model is providing the full answer. It is however presented as the full answer.

“(…) At the end of the paper, we added a short discussion on this. “

Indeed I see in the paper a small discussion on the internal variability in the CESM model (lines 353- 363). However, I do not really see the answer to the reviewers’ question, namely: what is the effect of the use of ‘only’ 1 model while in previous work multiple models were analysed? I know this is difficult, but I agree with the reviewer it needs to be addressed.

In the revision, we tried to better frame this discussion and pointed out that by using a multi-model ensemble, the sampled uncertainty in projected SLR will be the combination of the internal climate variability and the differences in model physics and configuration. Potentially, the latter may have a larger contribution. Therefore, to better isolate the effect of the internal variability on projected SLR, a single model with a large ensemble simulation could be a better choice. Of course, if large ensemble simulations were done using multi-models, the characteristics of the internal climate variability and its impact on projected SLR could be evaluated separately within each model, then a comparison whether a similar impact of internal climate variability on projected SLR is found and how different model physics and configurations impact internal climate variability can be investigated.

The discussion has changed to “Lastly, if multi-models are used, the effect of the internal variability on regional SLR will be severely contaminated by the differences in model physics and the horizontal-vertical configuration. This could result in larger uncertainty in projected SLR due to the combined effect of the internal climate variability and the model differences. Thus to better isolate the influence of the internal climate variability on project SLR, a single model with large ensemble simulation, as is done within this study, is a better choice. If large ensembles become available from different modeling groups, this study could be repeated with each ensemble and a comparison of SLR due to internal variability among the models can be conducted.”

discussion of model deficiencies (point 5)

5. To provide specific sea level rise values for different cities is problematic if only the thermal expansion of a coarse resolution climate model is provided. The spatial precision that is suggested by selection of a city is not appropriate when the numbers provided (1) are from a coarse resolution model, (2) only cover 40% of the currently observed sea level rise and (3) do not include potential contributions for example from tectonic uplift which can be as strong as the regional sea level or the global sea level rise.

This is a reasonable comment. However, as we have stated in our manuscript, the SLR discussed here includes both SLR due to thermal expansion of seawater and the dynamic sea level. If only the thermosteric sea level is included, it does not have a regional pattern because it was added onto the dynamic sea level as a global mean number. Regional SLR differences are due only to the dynamic effect (the changes of the ocean currents and dynamics due to changes in wind and buoyancy forcings) and can add to or subtract from the global mean regardless of what the definition of global mean is; therefore, the patterns of regional sea level under different future scenarios are important for regional planning in coastal areas This is clearly stated in our original manuscript.

I agree with the reviewer that using 'city' values can be problematic. The authors have in this new manuscript added a clarification to the methods section that the city values are taken from the nearest grid point. However, as I mentioned in my re-review, the sentence 'which may minimize the actual SLR variability' is a bit awkward and I would suggest to change it.

In the light of Rev#2's comments, I would strongly suggest to put some more explanation on why there could be differences between the 'open ocean' and the actual coastal change. E.g. missing processes, missing topography, propagation of steric signals to the coast. It deserves a bit more discussion than the current half sentence, as there is a 'danger' that people will take these values literally!

Additionally, the SLR noted for specific cities are actually part of a larger, regional response of SLR because it is driven by large scale climate variability (like the NAO, PDO, etc.); therefore, we could make generalizations about specific cities based on these larger patterns even if we had not done the analysis at specific grid points.

I would suggest to actually put this in the text, that would be very helpful.

Thanks for this comment. We have modified the text as "To link the potential impact of the SLR to human societies, here we use the SLR for selected global cities as shown in Supplementary Figure 1 as examples. The SLR for these cities is defined as the SLR of the closest ocean grid point, which may underestimate the actual SLR variability due the unresolved physical processes, such as ocean eddies and the detailed shape of coastlines. Additionally, the SLR noted for specific cities in this work is actually part of a larger, regional response of SLR to changes in the external forcing because the SLR is driven by large scale climate variability (like the NAO, PDO, etc.); therefore, we can make generalizations about specific cities based on these larger patterns even if we had not done the analysis at specific grid points. In reality, the coastal SLR is not only controlled by large scale climate patterns, but also affected by local winds on a spatial scale of tens kilometers in association with the ocean topography and shape of the coastlines (may also be affected by changes of tides in different timescales). If these processes were included in our simulations, it certainly would induce larger SLR variability in these coastal cities.

the level of the discussion of results (point 6)

6. The results of the simulation are generally just reported and not explained (for example by ocean circulation changes).

In general, we have two major parts in our paper. One is to show the sea level rise on global and regional scales and the uncertainty of the sea level rise; the other part is the explanation of how the internal variability affects the sea level change in terms of the dynamic sea level change. In our revision, we have tried to do a better job of explaining how the sea level rise patterns are related to the internal variability trends, and how the internal variability affects the sea level rise uncertainty.

I agree with Rev#2 that this was the case in the original manuscript. I think that in the new version the focus is much more in the internal climate processes and how they affect dynamic sea level, which is the new & noteworthy bit of the paper.

Thanks!